

# Linking variations in sea spray aerosol particle hygroscopicity to composition during two microcosm experiments

Sara D. Forestieri[1], Gavin C. Cornwell[2], Taylor M. Helgestad[1], Kathryn A. Moore[2], Christopher Lee[2], Gordon A. Novak[3], Camille M. Sultana[2], Xiaofei Wang[2], Timothy H. Bertram[2,3], Kimberly

A. Prather[2,4], Christopher D. Cappa[1,*]

[1]Department of Civil and Environmental Engineering, University of California, Davis, CA 95616
[2] Department of Chemistry and Biochemistry, University of California, San Diego, La Jolla, CA 92093
[3] Department of Chemistry, University of Wisconsin, Madison, WI 53706
[4] Scripps Institution of Oceanography, 9500 Gilman Drive, La Jolla, CA 92093

[*] Correspondence to: E-mail: cdcappa@ucdavis.edu

**Abstract.** The extent to which water uptake influences the light scattering ability of marine sea spray aerosol (SSA) particles depends critically on SSA chemical composition. The organic fraction of SSA can increase during phytoplankton blooms, decreasing the salt content and

therefore the hygroscopicity of the particles. In this study, subsaturated hygroscopic growth factors at 85% relative humidity ($GF$(85%)) of SSA particles were quantified during two induced phytoplankton blooms in marine aerosol reference tanks (MARTs). One MART was illuminated with fluorescent lights and the other was illuminated with sunlight, referred to as the "indoor" and "outdoor" MARTs, respectively. $GF$(85%) values for SSA particles were derived from

measurements of light scattering and particle size distributions, concurrently with online single particle and bulk aerosol composition measurements. During both microcosm experiments, the observed bulk average $GF$(85%) values were depressed substantially relative to pure, inorganic sea salt, by 10 to 19%, with a one (indoor MART) and six (outdoor MART) day lag between $GF$(85%) depression and the peak chlorophyll-a concentrations. The fraction of organic-

containing SSA particles generally increased after the peak of the phytoplankton blooms. The $GF$(85%) values were inversely correlated with the fraction of particles containing organic or other biological markers. This indicates these particles were less hygroscopic than the particles identified as predominately sea salt containing and demonstrates a clear relationship between SSA particle composition and the sensitivity of light scattering to variations in relative humidity. The

implications of these observations to the direct climate effects of SSA particles are discussed.





## 1    Introduction

Aerosols impact climate directly by scattering and absorbing solar radiation and indirectly by modifying cloud properties (IPCC, 2013). Sea spray aerosol (SSA) particles are a major source of natural aerosols to the atmosphere and dominate the pre-industrial clear-sky direct radiative effects

over the ocean (Haywood et al., 1999). Breaking waves in the ocean entrain air into seawater leading to the formation of bubbles, which burst at the ocean's surface, producing SSA particles (Lewis and Schwartz, 2004). The climate impacts of SSA particles depend critically on their composition, shape, and size (Carslaw et al., 2013; Pilinis et al., 1995). Given typical size distributions for particles in the marine boundary layer, both the submicron (300-1000 nm) and

supermicron (~1-5 μm) size ranges contribute to light scattering, with the relative contributions varying depending on location and conditions (Kleefeld et al., 2002). Under humidified conditions, the size of SSA particles is modified through water uptake and loss, which are a strong function of chemical composition (Saxena et al., 1995). The overall mass of SSA particles is dominated by sodium chloride and other inorganic ions, but organic compounds can also contribute substantially

to the total mass, especially in the submicron size regime (Facchini et al., 2008; Keene et al., 2007; O'Dowd et al., 2004). Larger organic matter-to-salt ratios occur in the submicron mode through the formation of film drops, since surface-active organics can become enriched in the thin film prior to bubble bursting (Skop et al., 1994; Stefan and Szeri, 1999; Tseng et al., 1992), but it has also been shown that supermicron particles can also contain organic and biological markers (Quinn

et al., 2015). Since organic compounds are universally less hygroscopic than inorganic sea salt (Petters and Kreidenweis, 2007), their transfer to SSA will lead to less water uptake and, thus, less scattering than in the case of pure inorganic sea salt particles of the same size.

Previous studies have linked the suppression of water uptake of ambient SSA particles to increasing fractions of marine-derived organic matter in the ambient atmosphere (Vaishya et al.,

2013; Ovadnevaite et al., 2011; Lawler et al., 2014; Hegg et al., 2008; Zhang et al., 2014). Phytoplankton blooms lead to chemical changes in seawater and serve as a source of particulate and dissolved organic carbon (DOC) to the system, which is then processed by other micro-organisms as part of the microbial loop (Pomeroy et al., 2007). These chemical and biological changes in the seawater can impact SSA particle composition (Prather et al., 2013; Lee et al., 2015;

O'Dowd et al., 2004). The organic fraction of SSA particles has been correlated to metrics for high





biological activity, such as chlorophyll-a, in some studies (O'Dowd et al., 2004; Facchini et al., 2008), but not others (Quinn et al., 2014). Some lab studies have observed small depressions in water uptake by SSA particles produced from natural seawater relative to synthetic, inorganic seawater (Sellegri et al., 2008; Modini et al., 2010; Fuentes et al., 2011; Park et al., 2014).

However, all of these studies have focused on particles smaller than ~150 nm, for which variations in composition have bigger impacts on cloud condensation nuclei concentrations (Dusek et al., 2006; Farmer et al., 2015). Substantially less is understood about the connections between seawater composition and the water uptake properties of the larger submicron particles that contribute more to light scattering. The connection between biological and chemical characteristics

of seawater and the resulting SSA particle composition, and consequently hygroscopicity, has therefore not been fully established.

To better understand the connection between SSA particle composition and water uptake as it relates to light scattering in particular, two microcosm experiments were conducted in July 2014 as part of IMPACTS (Investigation into Marine Particle Chemistry and Transfer Science). Through

the addition of light and nutrients, phytoplankton blooms were induced in natural seawater. Marine aerosol reference tanks (MARTs) were used to produce SSA particles via intermittent plunging of a sheet of water, which reproduces the bubble size distribution of whitecaps in the ocean (Lee et al., 2015; Stokes et al., 2013). Studying the SSA particles produced during these microcosm studies can provide insights into the linkage between hygroscopicity of nascent SSA particles and ocean

biology in an environment that is isolated from anthropogenic influence or background particles. The simultaneous measurement of submicron SSA particle water uptake and of particle composition here demonstrate that variations in seawater biology and composition influence water uptake by SSA particles.

## 2 Methods

### 25 2.1 MART Description and Operation

Two separate experiments were conducted during July 2014 utilizing MARTs. Detailed information on the performance and operation of MARTs can be found in Stokes et al. (2013) and only a brief description will be provided here. SSA particles were generated in an enclosed 210 L acrylic tank via an intermittent plunging sheet of water operated on a computer-controlled 4-



seconds-on, 4-seconds-off cycle to allow for surface foam evolution and dissipation close to what would be observed for natural whitecaps in the ocean. The plunging cycle creates a constant, reproducible concentration of nascent SSA particles in the 90 L headspace, with SSA particle size distributions produced in the MARTs generally consistent with those observed for SSA particles

from breaking waves in the ocean (Stokes et al., 2013). SSA particles sampled from the MARTs are primary, since the average residence time in the MARTs is much shorter than the time scale required for secondary processing of SSA particles (e.g. heterogeneous gas-phase reactions) (Lee et al., 2015). SSA particles from the headspace were sampled periodically each day by instruments that characterized SSA particle size distributions, composition, and optical and hygroscopic

properties. SSA were sampled from the MART headspace and transported through an approximately 2 m long line of 3/8 in. conductive tubing into a laminar flow manifold from which the instruments sampled. Due to the limited headspace volume and flow restrictions, not all instruments in this study could sample simultaneously. During individual sampling periods, only a subset of the full instrument suite sampled from the MARTs. The flow rate of zero air going into

the MART and the flow being pulled from the MART by the instruments, as well as sampling times for each group of instruments, are provided in Table 1. The flow into the MART was always 1 LPM greater than the combined instrument pull to ensure positive pressure in the headspace, which eliminated possibility of sampling of room air. The shape of the measured particle size distributions were relatively independent of flow rate for the range of flow rates considered while

sampling from the MARTs, although the residence time of individual particles decreases as the flow rate increases (Stokes et al., 2013). The air pushed into the MART was produced by a zero air generator (Sabio Instruments, Model 1001), with air flow controlled by a mass flow controller. The excess flow was released through a vent on the MART. The different flow rates through the MART and through the sample tubing for each sampling configuration led to some differences in

the size distribution sampled by the downstream instrumentation.

The procedure for inducing phytoplankton blooms inside of the MARTs will be briefly described here; further general details can be found in Lee et al. (2015). The MARTs were filled with ~120 L of seawater each collected from the SIO pier in La Jolla, California, USA (32°51′56.8"N: 117° 15′38.48"W). Debris and zooplankton were filtered out of the seawater with

50 μm mesh. Phytoplankton growth was induced by the addition of Guillard's f medium (a diatom growth medium) and $Na_2SiO_3$ solutions, both of which were diluted by a factor of 2, and exposure





to artificial or natural light. The two independent MART experiments will be referred to according to the MART location during the growth phase, either "indoor" or "outdoor". The indoor MART was illuminated using 5700 K full spectrum lights, while the outdoor MART was illuminated with sunlight. A key difference between these two experiments is the intensity of the photosynthetically

active radiation (PAR) during growth. The PAR was much greater for the outdoor MART compared to the indoor MART (PAR~1000-1500 $\mu$E m$^{-2}$ s$^{-1}$ (Bouvet et al., 2002) versus ~70 $\mu$E m$^{-2}$ s$^{-1}$), which resulted in a much larger peak Chl-a concentration for the outdoor MART, 51 $\mu$g L$^{-1}$ (outdoor) versus 10 $\mu$g L$^{-1}$ (indoor). An additional difference between the two mesocosms was that the seawater was collected on different days, 8 July for the indoor MART and 19 July for the

outdoor MART.  The conditions of the seawater at time of collection are detailed in Table 2.

On 9 July, particles from the indoor MART were sampled immediately following nutrient addition. Further sampling was delayed until Chlorophyll-a (Chl-a) concentrations exceeded 12 $\mu$g L$^{-1}$, which occurred seven days after nutrient addition. (The same seawater as used in the indoor MART was added to a separate MART and sampled immediately after collection and before

nutrient addition. However, the resulting particle size distribution from this MART differed substantially from those measured from the indoor MART, with a much greater contribution of large particles. Thus, the measurements from this separate MART are not directly comparable to the measurements from the indoor MART and are not considered further.)  The outdoor MART was only sampled after Chl-a concentrations exceeded 12 $\mu$g L$^{-1}$, which occurred three days after

nutrient addition. This delay in sampling from when the water and nutrients were first added to the MARTs is necessary because the plunging process can lead to lysis of the phytoplankton cells during this vulnerable growth period, which will inhibit phytoplankton growth (Lee et al., 2015). Ultimately, SSA from the initially collected water from the indoor MART was sampled on 9 July, and subsequent sampling commenced periodically from 19 July through 31 July, i.e. beginning 11

days after the water was collected. Sampling from the outdoor MART did not commence until 22 July, continuing through 1 August. During the growth period and the off-sampling periods, air was gently bubbled through the tank to provide aeration. Sampling from the MARTs was performed daily once the threshold chlorophyll-a concentrations were reached.



## 2.2 Instrumentation

A variety of online and offline measurements were made to characterize water composition and particles generated within the MARTs. A general sampling schematic is shown in Figure 1 and a list of the instrumentation used is given in Table 1. As only a limited number of instruments were able to sample concurrently from the MART due to flow limitations, the individual sampling configurations (i.e. groupings of instruments sampling at the same time) are indicated; three specific instrument groupings are considered. The sampling times of each group relative to Group 1 are listed in Table 1. (The specific timing was dictated by the broader goals of IMPACTS.) A general description of the key instrumentation used as part of this study is provided below. Group 1 sampled for 1.5 h, group 2 sampled for 2 h, and Group 3 for sampled 1 h each day that sampling was conducted.

### 2.2.1 Online particle measurements

Size distributions for dried particles (RH < 20%) were measured with a scanning electrical mobility sizer (SEMS; BMI; model 2002), and an aerodynamic particle sizer (APS; TSI Inc.; Model 3321). The SEMS combines a differential mobility analyzer (DMA) and a mixing condensation particle counter (MCPC) to characterize particles according to their mobility diameter ($d_{p,m}$). The APS characterizes particles according to their aerodynamic diameter ($d_{p,a}$). The SEMS characterized particles over the range 10 nm < $d_{p,m}$ < 1900 nm and the APS over the range 0.7 μm < $d_{p,a}$ < 20 μm. The SEMS size distributions were corrected for the influence of multiply charged particles using software provided by the manufacturers. No diffusion correction was performed, which has negligible influence on this study because the smallest particles, which are sensitive to diffusion corrections, contribute negligibly to the observed scattering. The APS had a time resolution of 1 minute, while the SEMS had a time resolution of 5 minutes and the APS distributions were accordingly averaged to 5 minutes to facilitate generation of a merged size distribution. The SEMS and APS distributions were merged using the SEMS distribution up to 1 μm and the ($d_{p,m}$ equivalent) APS distribution at larger diameters. The APS $d_{p,a}$ values were converted to mobility equivalent values assuming a particle density of 1.8 g cm$^{-3}$.

The hygroscopicity of the SSA particles was characterized through simultaneous measurement of light extinction coefficients ($b_{ext}$) for particles that were either dried to RH < 20% ("dry") or humidified to RH ~85% ("wet") using the UC Davis cavity ringdown spectrometer





(CRD) (Langridge et al., 2011; Cappa et al., 2012). Light absorption by the SSA particles was negligible, and thus extinction is equal to scattering, i.e. $b_{ext} = b_{sca}$. The dry particle measurements were made at wavelengths of 532 nm and 405 nm, while the wet particle measurements were made only at 532 nm. It should be noted that the humidified particle stream was generated without first

drying the particles, and thus it is unlikely that the sampled particles had effloresced. Humidification was achieved by passing the particles through a Nafion humidifier (Permapure, MD-110-12) while drying was achieved by passing the particles through a diffusion denuder filled with Drierite.  Both the humidifier and drier were oriented vertically to prevent differential losses due to sedimentation, which could bias the measurements. The fundamental performance of the

CRD method for wet particles is the same for dry particles, but variations and uncertainty in the relative humidity (RH) contribute to the uncertainty in the measured $b_{ext}$. The RH for the humidified channel varied between 80-87% due to challenges in maintaining a constant temperature in the open-air Scripps Hydraulics Lab; these variations are accounted for in the analysis as described below. The RH of the air was measured directly in the CRD cells using RH

probes (Vaisala, HMP50) that were calibrated against saturated salt solutions. The wet (high RH) and dry (low RH) particle measurements are combined to provide a characterization of the extent of water uptake at a given RH, which causes particles to grow through the parameter $f$(RH), where:

$$f(RH) = \frac{b_{ext}(RH_{high})}{b_{ext}(RH_{low})} = \frac{b_{sca}(RH_{high})}{b_{sca}(RH_{low})} \qquad (1).$$

The parameter $f$(RH) is RH-specific, and is most appropriate when $RH_{low}$ is sufficiently low that there is little, if any particle-phase water. The accuracy of the $f$(RH) measurements, as well as the conversion to equivalent growth factors ($GF$, Section 2.2.3), were tested through measurements made using sodium chloride and ammonium sulfate particles that were generated using an

atomizer. The CRD (and SEMS) alternated between sampling behind a PM$_{2.5}$ cyclone and with no explicit size cut (referred to as PM$_{all}$) every ten minutes to try and determine $f$(RH) and $GF$ values separately for smaller and larger particles. However, since the measured size distributions indicate minimal contributions from particles with $d_{p,a} > 2.5$ µm, the PM$_{2.5}$ and PM$_{all}$ measurements will generally be considered together.





An aerosol time of flight mass spectrometer (ATOFMS) (Gard et al., 1998; Pratt et al., 2009) was used to characterize the composition of individual dried SSA particles with vacuum aerodynamic diameters ($d_{va}$) from ~300 nm to 3 μm, with the highest transmission and sampling of particles with $d_{va}$ ~1-2 μm (Wang et al., 2015). The ATOFMS single particle spectra have been

analyzed using a statistical clustering algorithm (ART-2a) that groups particles with similar spectra together (Zhao et al., 2008). Six distinct particle types were identified here and are classified as: sea salt (SS), salt mixed with organic carbon (SSOC), predominately OC containing (OC), containing a large Fe peak (Fe) and containing a large Mg Peak (Mg) (Lee et al., 2015; Sultana et al., In Prep.; Wang et al., 2015). A campaign-average spectrum for each particle type is shown in

Figure S1. Each type is reported as a fraction of the total particles sampled for the following vacuum aerodynamic diameter size bins individually: 0.5-1 μm, 1-1.5 μm, 1.5-2 μm, 2-2.5 μm, referred to as the 0.75 μm, 1.25 μm, 1.75 μm and 2.25 μm bins, respectively. These correspond approximately to mobility diameters of 420 nm, 690 nm, 970 nm and 1250 nm, respectively, assuming a density of 1.8 g cm$^{-3}$ and spherical particles. The combination of the aerodynamic lens

transmission and the input particle size distribution determines the particular weighting of the average fractions of the ATOFMS particle types (see Figure S2); in this study, the weighted-average corresponds approximately to particles with $d_{va} = 1.5$ μm and, unless otherwise specified, the results are for the sampling-weighted average.

An Aerodyne high resolution time-of-flight aerosol mass spectrometer (HR-ToF-AMS,

henceforth AMS) quantified mass concentrations of non-refractory (NR) components of dried SSA particles, in particular NR organic matter (NR-OM) but also other non-refractory (NR-PM) components (Canagaratna et al., 2007). NR-PM species are defined as those that volatilize at ~600 °C on a time scale of a few seconds under vacuum ($10^{-4}$ torr) conditions. No cyclone was used in front of the AMS, and thus the size range of sampled SSA particles was determined by the size-

dependent transmission of the aerodynamic lens, which nominally allowed for quantitative sampling of particles with $d_{va}$ between 90 nm and 700 nm (50% cut points at ~40 nm and ~1 micron), although some fraction of even larger particles were characterized (Wang et al., 2015). The AMS data were analyzed using the SQUIRREL toolkit. The instrument collection efficiency ($CE$) was assumed to be unity, and thus NR-OM values are likely underestimated for particles

containing refractory sea salt (Frossard et al., 2014). The NR-OM fraction of total sampled PM was estimated by normalizing the NR-OM mass concentrations by PM$_1$ concentrations determined





from integration of the SEMS particle size distributions using an assumed density of 1.8 g cm$^{-3}$. Since a $d_{va}$ of 1 µm corresponds approximately to a $d_{p,m}$ = 560 nm, the use of the SEMS size distribution is appropriate and the derived NR-OM fractions can be considered reflective of the submicron SSA composition. It is important to note that while the temporal trends of the AMS

5    NR-OM/PM$_1$ fractions are likely reflective of the general behavior, the absolute values are likely too low because NR-OM associated with particles containing high sea salt fractions is not vaporized efficiently by the AMS due to the refractory nature of sea salt and to the susceptibility of SSA particles to particle "bounce" in the AMS, i.e. have $CE$ values <1 (Frossard et al., 2014). The high resolution mass spectra were analyzed using the PIKA toolkit to determine O/C atomic

10   ratios for the NR-OM components.

The $f$(RH) values measured using the CRD instrument have been converted to physical growth factors ($GF$s), defined as:

$$GF(RH) = \frac{d_{\mathrm{p}}(\mathrm{RH_{high}})}{d_{\mathrm{p}}(\mathrm{RH_{low}})} \tag{2}$$

where $d_{\mathrm{p}}$ is the geometric particle diameter, which is equivalent to $d_{p,m}$ for spherical particles. Unlike $f$(RH), $GF$ values are independent of the dry particle size, and thus only depend on composition. The optical closure technique uses spherical particle Mie theory calculations and the measured size distributions and $f$(RH) values to derive equivalent $GF$(RH) values. This

20   methodology is described in detail in Zhang et al. (2014). In brief, the dry scattering is first calculated from the measured dry particle size distribution assuming a refractive index of 1.55 (the refractive index for NaCl), as:

$$b_{sca} = \int \sigma_{sca}(d_{p,m}) \cdot \frac{dN}{d \log d_{p,m}} d \log d_{p,m} \tag{3}$$

where $\sigma_{sca}$ is the size-dependent scattering cross section and $d\mathrm{N}/d\log d_{p,m}$ is the number-weighted size distribution. Then, each diameter for the dry distribution is multiplied by a trial value for



*GF*(RH), the refractive index of the particles is adjusted to account for the resulting volume fraction of water, and the scattering by the resulting "wet" distribution is calculated, from which a theoretical *f*(RH) value is determined. The calculated *f*(RH) is compared to the observed *f*(RH), and if the two do not agree to within 0.01 the trial *GF*(RH) is increased until closure is obtained.

As the RH of the humidified channel was not perfectly constant during measurements, the derived individual *GF*(RH) values have been adjusted to 85% by using Equation 4:

$$\frac{RH}{\exp\left(\frac{A}{d_d * GF(RH)}\right)} = \frac{GF(RH)^3 - 1}{GF(RH)^3 - (1 - \kappa)} \tag{4}$$

where $A$ is a constant, RH is relative humidity, $d_d$ is the dry particle diameter and $\kappa$ is the effective hygroscopicity parameter, which is assumed to be RH-independent (Petters and Kreidenweis, 2007). Here, the $d_d$ values used are the optically-weighted median diameters, which are calculated by integrating the concentration-weighted size-dependent cross-sections ($\sigma_{sca}(d_p)$). *GF*(85%) values were determined by first calculating $\kappa$ based on the measured *GF*(RH) and then

recalculating the *GF* at 85% RH.

     The accuracy of this optical closure method, as well as of the initial *f*(RH) measurements, was assessed by comparing the *GF*(85%) values determined for NaCl and $(NH_4)_2SO_4$ test particles, for which *GF*(85%) values are known. The measured *GF*(85%) for NaCl was 2.09 +/- 0.03 and for $(NH_4)_2SO_4$ was 1.59 +/- 0.05, which compare very well with literature values of ~2.1 for NaCl

(Cruz and Pandis, 2000; Laskina et al., 2015; Hansson et al., 1998) and ~1.55 for ammonium sulfate (Laskina et al., 2015; Wise et al., 2003). (The reported experimental uncertainties are 1σ standard deviations over each measurement period.)

     *GF* calculations for PM$_{all}$ utilized a combined size distribution from the SEMS and the APS, with the merge point at a $d_{p,m}$ = 1000 nm. The APS sampled at a separate time from the CRD (see

Table 1). The CRD set-up also required dilution due to the 3 LPM required for the cyclone and a total pull of ~6.3 LPM. Therefore, a dilution correction was applied to the APS distributions to account for the different sampling scheme. Although this adjustment adds some uncertainty to the





PM$_\text{all}$ size distributions, the concentrations at larger sizes were very small and thus had minimal influence on the derived $GF$s. For the PM$_{2.5}$ sampling periods, only SEMS distributions were used.

## 3   Results

### 3.1   Size Distributions and Dry Particle Optical Closure

The daily and study average merged size distributions for each MART are shown in Figure 2A (indoor MART) and Figure 2B (outdoor MART). The day-to-day variations in the size distributions were generally small. The average SSA particle number-weighted size distributions from both MARTs peaked around $d_{\text{p,m}}$ = 100 nm and were relatively broad. The observed concentration of supermicron particles ($d_{\text{p,m}} > 1000$ nm) was somewhat lower than that previously reported from a MART (Stokes et al., 2013) and likely reflects greater gravitational losses of supermicron particles in the long sampling line used here (Figure S3). Since the hygroscopicity measurements discussed in this study are based on measurements made using polydisperse distributions, it is useful to determine the effective, scattering-weighted particle diameters that characterize the MART size distributions. The study average integrated scattering for each MART was calculated from Mie theory using the observed dry particle size distributions (Figure 2C). The diameters at which 50% of the total scattering occurs were 570 nm for the outdoor MART and 530 nm for the indoor MART and particles with $d_{\text{p,m}} > 1000$ nm contributed <10% of the total scattering in both MARTs,, indicating that the derived $GF(85\%)$ values for these two experiments are most sensitive to submicron particles with $d_{\text{p,m}}$ values between about 400 nm and 800 nm.

The extent of agreement between the observed $b_{\text{sca}}$ for dry particles and the values calculated from Mie theory using measured size distributions (Equation 3) has been assessed (Figure S4). The calculated $b_{\text{sca}}$ are ~15% lower than the observed $b_{\text{sca}}$ for both PM$_\text{all}$ and PM$_{2.5}$, which is outside the combined uncertainty for the CRD and size distribution measurements (which is ~11% from error propagation). Some of the difference may result from differential losses between or within the sizing instruments and the CRD, although this seems generally unlikely to explain the entire difference, as losses of particles in the submicron range should be small. There is greater scatter in the PM$_\text{all}$ light scattering comparison than there is from the PM$_{2.5}$ comparison, which likely results both from the APS measurements being made at a different time than the CRD and SEMS



measurements and the need for dilution correction. Some of the difference between the observed and calculated $b_{sca}$ may be attributable to the assumption of spherical particles in the calculations, although similar closure was obtained (within 16%) between observed and calculated $b_{sca}$ for atomized NaCl, suggesting that this is unlikely to explain the difference. It is possible that the

diameters measured by the SEMS may have been too small. If measured diameters are increased by 8%, then a 1:1 agreement between the measured and calculated extinction values is obtained. However, tests conducted during the study in which a 2$^{nd}$ DMA was used to size-select monodisperse particles in the range 100-300 nm indicated agreement between the instruments to within 1%. Additional tests after the study using 220 nm monodisperse polystyrene latex spheres

(PSLs) demonstrated the SEMS sizing was good to better than 1%, suggesting that sizing inaccuracies cannot explain the difference absent some fundamental problem with the data inversion procedure for size distributions (Lopez-Yglesias et al., 2014), which seems unlikely. Uncertainty in the assumed RI value for the dry particles may explain a small fraction (<5%) of the difference. Additionally, if the dry particles had retained some water in the CRD but not the

SEMS, then the observed $b_{sca}$ would be larger than the calculated value. However, the RH in the CRD dry channel is much lower than the efflorescence RH for NaCl (~45% (Biskos et al., 2006)), and thus it seems unlikely that residual water would have contributed substantially to the difference. Regardless of the explicit reason for the difference in calculated and observed absolute values of $b_{sca}$, since the calculation of $f$(RH) depends on the ratio between the $b_{sca}$ for wet and dry

particles, such absolute differences do not strongly affect the retrieval of $GF$(85%) values. We have tested the sensitivity of the retrieval method to an 8% increase in the particle diameters. The retrieved $GF$ values are increased by a marginal amount (0.015-0.03) when the diameters are increased, and thus such potential sizing uncertainty does not affect the main conclusions presented here.

**3.1   Indoor MART**

The temporal variation in Chl-a concentrations, the derived $GF$(85%) and various particle composition metrics are shown in Figure 3 for the indoor MART. As has been previously observed in microcosm experiments, the measured Chl-a time series exhibits a distinct peak (Lee et al., 2015), which in this case occurred on 16 July at a value of 10 μg L$^{-1}$. This Chl-a concentration is

around the upper end of values observed for large phytoplankton blooms observed in the oceans,



in particular near coastal regions (O'Reilly et al., 1998). After the peak the Chl-a concentration dropped relatively quickly to around 1.5 µg L$^{-1}$ (15% of the peak) and then eventually to ~1.4 µg L$^{-1}$ (14% of the peak). The DOC concentrations varied from 240 to 350 µM C, increasing rapidly when the Chl-a concentration peaked and then staying relatively constant around 320 µM (Figure

S5A). The peak DOC range is somewhat larger than values typically observed for blooms in the ocean, which are only ~130 - 250 µM C (Kirchman et al., 1991; Norrman et al., 1995). The temporal variation in heterotrophic bacteria concentration was similar to that for DOC, and heterotrophic bacteria concentrations ranged from ~1 x 10$^6$ to 1.2 x 10$^7$ mL$^{-1}$ (Figure S5A).

The $GF$(85%) values determined for the indoor MART ranged from 1.79 to 1.9 and exhibited

distinct temporal variations, decreasing from $1.88 \pm 0.04$ on 16 July, just as the Chl-a peaked, to a minimum range of $1.79 \pm 0.03$ to $1.80 \pm 0.01$ from 17 July to 18 July when the Chl-a concentration dropped to $3.41 \pm 1.89$ µg L$^{-1}$, and then recovering back to $1.90 \pm 0.03$ on 7/20 (Figure 3A). The range of these values is 14-19% lower than the value of ~2.2 for pure (inorganic) sea salt (Ming and Russell, 2001; Hansson et al., 1998), which is primarily NaCl. There is a one day lag between

the peak in Chl-a and the (temporary) depression in $GF$(85%).

With the exception of the measurements made just after nutrient addition on 9 July, the temporal variations in $GF$(85%) showed a strong, positive correlation ($R^2 = 0.83$) with the relative fraction of SS-type particles ($f_{SS}$) from the ATOFMS (Figure 3A). This key observation suggests that the predominant non-SS particle types during this period (SSOC, Mg and Other, listed in order

of decreasing fractional abundance) are, on average, less hygroscopic than the SS-type particles. However, very different behavior was observed on 9 July, and if the measurements from this day are included the correlation between $GF$(85%) and $f_{SS}$ decreases dramatically ($R^2 \sim 0$). This is because on this day only the Fe-type particles comprised a substantial fraction of the total particles ($f_{Fe} = 0.32 \pm 0.01$); on all other days the Fe-type particle fraction was negligible. Consequently,

the $f_{SS}$ on 9 July was much smaller, only 0.32. The sum of $f_{SS}$ and $f_{Fe}$ on 9 July is similar to the $f_{SS}$ values on the other sampling days. The $GF$(85%) values, including 9 July, therefore show a reasonable correlation with $f_{SS} + f_{Fe}$ ($R^2 = 0.49$). This suggests that the hygroscopicity of the Fe-type particles is similar to that of the $f_{SS}$ particles. However, the observed $GF$(85%) value on 9 July is somewhat smaller than expected given the correlation observed on the other days and the

$f_{SS} + f_{Fe}$ value on this day. This therefore suggests that the Fe-type particles are more hygroscopic



than the other non-SS-type particles but that they are not as hygroscopic as sea salt. These Fe-type particles result from nutrient addition at the beginning of the experiment, as the Guillard's f medium contains iron citrate (Guillard and Ryther, 1962). The Fe-type particles are characterized by their large $Fe^+$ peak, along with peaks corresponding to $Na^+$, $Cl^-$, $Mg^{2+}$ and, notably,

$PO_3^-$ (Figure S1). Accordingly, the Fe-type particles may be composed of substantial amounts of sea salt with a clump of insoluble Fe-rich particulates, i.e. might be reasonably considered as SS+Fe-type particles (Sultana et al., In Prep.). Although the Fe-type particles clearly have an influence on the observed $GF(85\%)$ values when present with high relative abundance, their large fraction on 9 July is a result of nutrient addition and not biological changes in the seawater, and

thus the measurements on 9 July are excluded from subsequent analysis.

As noted above, the non-SS particle types (excluding the Fe type) are, on average, less hygroscopic than the SS-type particles. Correlations with individual other particle types were generally weaker, although the minimum $GF(85\%)$ values observed on 17 July-18 July are coincident with maxima in SSOC, Mg type and, to a lesser extent, "Other" particle type , which

suggests some sort of anti-correlation between $GF(85\%)$ and these particle types (Figure 3B). It should be recognized, however, that since the ATOFMS particle types are reported as relative fractions, an increase in one type will correspond to a decrease in all other types. As such, it should not be entirely surprising that some anti-correlation between the $GF(85\%)$ and the non-SS particle types is observed given the strong positive correlation between $GF(85\%)$ and the SS particle type.

Co-variations amongst the non-SS particle types may occur that limit the extent of correlation between $GF(85\%)$ and any individual non-SS particle type fraction. Regardless, the particularly strong positive correlation between the $GF(85\%)$ values and the $f_{SS}$ values (excluding 9 July) suggests that the $GF(85\%)$ values for the non-SS particle types must be similar. If this were not the case, i.e. if each non-SS particle type had a distinctly different $GF(85\%)$, then the specific

variations in the non-SS particle types should have led to loss of correlation with the SS-type particles.

There is also an increase in the $NR-OM/PM_1$ ratio from the AMS during the general period when the $GF(85\%)$ values are at their minimum, although the peak in NR-OM is somewhat sharper than either the dip in the $GF(85\%)$ or in $f_{SS}$. The AMS characterizes only non-refractory

components of particles that do not "bounce." Thus, the observed variations in the $NR-OM/PM_1$ ratio neither account for any potential variations in refractory organics, which depends on the type





of organics sampled (Frossard et al., 2014) nor for changes in the AMS collection efficiency. Thus, it is possible that there are additional variations in the particle composition that are not entirely captured by the observed $NR\text{-}OM/PM_1$ ratio, but that influence the $GF(85\%)$ to some extent. It is possible that these aspects of the AMS are responsible for the difference in the temporal variability

compared to $GF(85\%)$ and $f_{SS}$. The O:C ratio of the organics that are measured by the AMS had an average value of $0.25 \pm 0.05$ ($1\sigma$), which is similar to the value of $0.20 \pm 0.08$ reported by Frossard et al. (2014) for primary NR-OM that was generated from the open ocean using the "sea sweep" [Bates et al., 2011]. The O:C ratio of NR-OM in the indoor MART generally increased with time, from 0.17 to 0.30, but also exhibited a temporary decrease on 17 July, the day when the

$GF(85\%)$ and $f_{SS}$ both first dropped. Since O:C often correlates with hygroscopicity for organics (at least for multi-component mixtures), this behavior may indicate a general increase in the hygroscopicity of the NR-OM with time (Cappa et al., 2011; Massoli et al., 2010). However, since the hygroscopicity of organic aerosol with O:C values in this range has generally been found to be small, the observed variations in O:C may not have a noticeable impact on the overall behavior of

the $GF(85\%)$ values.

Under the assumption that the SS-type particles are effectively pure sea salt particles, the $GF(85\%)$ for these particles should be ~2.2. In contrast, the $GF(85\%)$ values for the non-SS particle types are not known *a priori*, although reasons for their being less hygroscopic than the SS particle type can be postulated based on their mass spectra (Sultana et al.; Lee et al., 2015;

Collins et al., 2013). The SSOC particle type is identified in large part by the presence of organic-containing peaks in the mass spectrum (in particular, $CN^-$ and $CNO^-$; see Figure S1) (Lee et al., 2015). Thus, it makes sense that an increase in the SSOC particle type fraction at the expense of the SS-type particle fraction would suppress $GF(85\%)$ values, as organic compounds are less hygroscopic than sea salt. The Mg-type particles are characterized by their large $Mg^+$ peak, but

also by large $Cl^-$ peaks and smaller $Ca^+$ and $K^+$ peaks (Figure S1). It may be that Mg-type particles are sea salt particles in which, during the drying process, NaCl has crystallized first, leaving a Mg-rich shell on the outside of the particles (Cziczo and Abbatt, 2000). If these particles then undergo incomplete ionization, the Mg signal may dominate the mass spectra. If this is the case, then the Mg-type particles should be about as hygroscopic as the SS particle type, which appears to run

counter to the observations. However, if these Mg-type particles are highly enriched in $MgCl_2$ over NaCl, then their hygroscopicity would be reduced since $MgCl_2$ is less hygroscopic than NaCl





(Cziczo and Abbatt, 2000). An alternative interpretation of the Mg-type is that they are whole or fragmented bacteria cells (Guasco et al., 2013) or possibly strongly associated with organic material such as in marine gels (Gaston et al., 2011), in which case their hygroscopicity would likely be lower than sea salt.

5        The explicit co-variation of the SS-type particle fraction and the $GF(85\%)$ values is shown in Figure 4. Assuming mixing of two distinct particle types, with respect to hygroscopicity, and assuming additivity of the individual particle type $GF$ values, the overall, effective $GF$ ($GF_{tot}$) can be estimated as:

$$GF_{obs} = f_{SS} \cdot GF_{SS} + (1 - f_{SS}) \cdot GF_{non-SS} \qquad (5)$$

where $f_{SS}$ is the SS-type particle fraction and a single $GF$ value is assumed for all non-SS particle types, as discussed above. The line connecting $GF_{SS}(85\%) = 2.2$ and $GF_{non-SS}(85\%) = 1.0$ provides the minimum value expected for any combination of SS and non-SS particles. Low $GF_{non-SS}(85\%)$

values (~1.0) have been observed for fatty acids (Vesna et al., 2008), which have been found in SSA particles in the atmosphere (Mochida et al., 2002) and were observed in SSA produced in a related mesocosm experiment (Wang et al., 2015; Cochran et al., Submitted). Values above this line indicate that the $GF$ of the non-SS particle type is, on average, greater than 1. Equation 5 was fit to the data shown in Figure 4 to determine an average value for $GF_{non-SS}(85\%)$ for the indoor

MART. The best-fit $GF_{non-SS}(85\%)$ was $1.39 \pm 0.03$ when the sampling-weighted average $f_{SS}$ values from each day were used.

One important issue to consider in assessing the quantitative nature of the derived $GF_{non-SS}(85\%)$ value is that the sampling-weighted number fractions used above to determine $f_{SS}$ do not necessarily have the same weighting with respect to particle size as do the $GF(85\%)$ values.

(For clarity, in the following discussion the sampling-weighted average number fractions will have "avg" appended as a subscript.) The scattering-weighted median diameter, relevant to the $GF(85\%)$ measurements, was $d_{p,m} = 530$ nm whereas the sampling-weighted median diameter of the ATOFMS was $d_{va} \sim 1.5$ μm ($d_{p,m} \sim 800$ nm). The observed $f_{SS}$ values generally decreased as particle size decreased (Figure S6). This suggests that the actual $f_{SS}$ values most relevant to the

$GF(85\%)$ measurements should be somewhat smaller than the $f_{SS,avg}$. To provide for better





correspondence between the size sensitivity of the $f_{SS}$ values and the $GF(85\%)$ values, the $f_{SS}$ values have been adjusted by multiplying each $f_{SS,avg}$ value by the campaign-average of the ratio $R_{SS} = f_{SS,0.75\mu m}/f_{SS,avg}$, where $f_{SS,0.75\mu m}$ is the $f_{SS}$ for the $d_{va} = 0.75$ µm ($d_{p,m} \sim 420$ nm) bin. The campaign-average $R_{SS} = 0.78 \pm 0.05$. The absolute values of these re-weighted $f_{SS}$ values (referred

to as $f_{SS*}$) are more representative of the size range relevant to the $GF$ measurements. The reason for using a campaign-average scaling factor rather than using the measured $f_{SS,0.75\mu m}$ values directly is that, although the $f_{SS,avg}$ and $f_{SS,0.75\mu m}$ exhibit generally good correspondence, the counting statistics of particles in the 0.75 µm bin was substantially smaller than that in the larger bins, making the individual $f_{SS,0.75\mu m}$ points less certain than the campaign average. By using a

campaign-average scaling factor, it is implicitly assumed that the actual variations in $f_{SS,0.75\mu m}$ are captured by $f_{SS,avg}$, which seems reasonable given the general constancy of the size distributions over the course of each of the microcosm experiments, c.f. Fig. 2.

The relationship between $GF(85\%)$ and $f_{SS*}$ is also shown in Fig. 4, along with a separate fit using Eqn. 5. The $GF_{non-SS}(85\%)$ derived using $f_{SS*}$ is $1.57 \pm 0.03$, larger than that obtained when

$f_{SS,avg}$ is used. Regardless of whether $f_{SS}$ or $f_{SS*}$ is used, it is apparent that the non-SS particles types have, on average, $GF(85\%)$ values that are consistent with their being a mixture of less hygroscopic organic and more hygroscopic inorganic (sea salt) components. If it is assumed that the organic fraction is completely non-hygrocopic ($GF(85\%) = 1.0$) and the inorganic fraction is pure sea-salt ($GF(85\%) = 2.2$), the range of derived $GF_{non-SS}(85\%)$ values implies an average

organic volume fraction ($\varepsilon_i$) of $0.56 - 0.88$ for these particle types if it is assumed that volume mixing rules apply (i.e. the Zdanovskii-Stokes-Robinson mixing rules (Stokes and Robinson, 1966)). Since the $f_{non-SS}$ values range from $0.53 - 0.74$, if it is assumed that the SS-type and non-SS particle types have similar size distributions, then the implied ensemble average $\varepsilon_i$ would be about $0.33 - 0.52$. If the organics comprising the non-SS particles are somewhat hygroscopic, then

the estimated $\varepsilon_i$ values would be larger. The range of estimated $\varepsilon_i$ values is in line with values observed in the ambient atmosphere during periods of relatively high biological activity (O'Dowd et al., 2004).



## 3.2 Outdoor MART

The temporal variation in Chl-a concentrations, the derived $GF(85\%)$ values and various particle composition metrics are shown in Figure 5 for the outdoor MART. Like the indoor MART, the Chl-a concentrations exhibited a characteristic rise and fall for the microcosm experiment.

However, the maximum Chl-a concentration was 51 µg L$^{-1}$, which 5 times higher than the indoor MART and likely due to greater PAR in the outdoor MART. Such high Chl-a concentrations are well above those typically observed in the ocean. However, the Chl-a concentration rapidly declined to 6 µg L$^{-1}$ two days after the peak and then continued to decrease over the next week to <1.5 µg L$^{-1}$. Both DOC and heterotrophic bacteria concentrations increased as the bloom

progressed until they stabilized around the point when Chl-a concentrations had returned approximately to their pre-bloom levels, with DOC concentrations ranging from 200 to 300 µM C and heterotrophic bacteria concentrations from 1 x 10$^6$ to a peak of 1.7 x 10$^7$ mL$^{-1}$ (Figure S5B).

The $GF(85\%)$ values ranged from a maximum of $1.99 \pm 0.03$ to a minimum of $1.78 \pm 0.04$, again lower than what would be expected for pure sea salt (by 10-19%). Unfortunately, no pre-

bloom measurements were possible for this experiment, with the first particle measurements made for all instruments when the Chl-a concentration was peaking. The smallest $GF(85\%)$ values were observed towards the end of the microcosm, when the Chl-a concentrations were at their lowest point (< 1.5 µg L$^{-1}$). The $GF(85\%)$ values exhibited two sequential decreases after the Chl-a peak, the first after 3 days and the second after 6 days. The $f_{SS,avg}$ values for the outdoor MART were

generally smaller than those observed for the indoor MART experiment and varied over a wider range (0.32 to 0.58 versus 0.55 to 0.6), most likely a reflection of the higher peak Chl-a concentration and greater overall biological activity, although interestingly the DOC concentrations reached higher levels in the indoor MART. Despite the smaller $f_{SS,avg}$ for the outdoor MART, the range of $GF(85\%)$ values was similar to the indoor MART, if not a bit higher.

Nonetheless, there is still a moderate, albeit weaker, positive correlation ($R^2 = 0.48$) between the $GF(85\%)$ values and $f_{SS,avg}$ (Figure 4 and Figure 5A). The two most abundant non-SS particle types were SSOC-type and Mg-type, with all other types contributing negligibly. The Mg-type particles exhibited a relatively constant fraction while the variation in the SSOC-type particle fraction ($f_{SSOC}$) was substantial. Consequently, most of the variation in $f_{SS,avg}$ is coupled with variation in

$f_{SSOC}$. With exception of the data point on 7/30 (Figure 5C), there was co-variation between AMS NR-OM/PM$_1$ and $GF(85\%)$, but the overall correlation was not quite as strong as the relationship





between the $f_{SS,avg}$ and $GF(85\%)$, the reasons for which are not entirely clear. As with the indoor MART, this could be indicative of some fundamental lack of relationship, or it could be a result of limitations of the AMS in terms of characterizing the suite of organic compounds that are contributing to variations in the $GF$ values.

The $f_{SS,avg}$ decreases gradually over a few days just after the peak in Chl-a, stabilizes, and then exhibits a relatively sharp drop six days after the bloom peak that persists for about two days before recovering (Figure 5A). This behavior in $f_{SS,avg}$ tracks neither the Chl-a nor DOC behavior, suggesting that perhaps biological processing, more so than absolute organic concentrations, is important for determining the predominant types of particles that are produced and, presumably,

the abundance of organic matter transferred into SSA particles (Rinaldi et al., 2013; Lee et al., 2015; Quinn et al., 2014). However, further experiments will be needed to confirm this hypothesis.

     Equation 5 has again been used to estimate $GF_{non-SS}(85\%)$ from the observed $GF(85\%)$ and $f_{SS,avg}$. Assuming $GF_{SS}(85\%) = 2.2$, the derived $GF_{non-SS}(85\%) = 1.60 \pm 0.03$ (Figure 4). If the size-adjusted $f_{SS*}$ is instead used, the derived $GF_{non-SS}(85\%)$ increases to $1.70 \pm 0.03$. (The

adjustment factor, $R_{SS}$, for the outdoor MART is $0.74 \pm 0.06$.) The $GF_{non-SS}(85\%)$ values using $f_{SS,*}$ is slightly larger than that obtained from the indoor MART, suggesting that the non-SS particles generated from the outdoor MART are perhaps slightly more hygroscopic than those from the indoor MART. There is one notable outlier from the general $GF(85\%)$ versus $f_{SS}$ relationship that occurred on 22 July, at the peak of the bloom, specifically with the observed $GF(85\%)$ being lower

than expected based on the general relationship (Figure 4). There is no clear chemical difference (as indicated by the ATOFMS particle fractions or AMS) of the particles on this date. It could be that there was some underlying chemical change in the nature of either the SSOC or Mg-type particles after this date that is not fully captured by the mass spectra analysis methods, but it is of note that the point on this date overlaps well with the indoor MART results. The estimated range

of average organic volume fractions of the non-SS particle types is 0.59-0.68, again assuming that the organics comprising non-SS particle types are non-hygroscopic while the inorganics behave like sea salt and that the ZSR mixing rule applies. Given the range of $f_{non-SS}$ values, this corresponds to an ensemble average $\varepsilon_i$ of $0.22 - 0.28$.





## 4    Implications and Conclusions

The two MART microcosm studies provide two case studies relating variations in the optically-weighted $GF(85\%)$ values and SSA particle composition. For both microcosms, clear depression of the $GF(85\%)$ values, relative to that for pure sea salt, occurred following the peak in Chl-a concentrations and upon the death of both phytoplankton blooms, but with differing time lags between peak Chl-a and the minimum $GF(85\%)$ between the experiments. The variations in $GF(85\%)$ showed relatively strong correlation with the number fraction of sea salt-type particles, as characterized by ATOFMS, with lower $GF(85\%)$ values corresponding to lower $f_{SS}$ values. This provides evidence that the non-SS particle types are, on average, less hygroscopic than the SS-type particles. The most abundant non-SS particle types identified here were sea salt mixed with organic carbon and Mg-containing. For a given microcosm experiment, the hygroscopicity of these different non-SS particle types was generally similar. However, the non-SS particle types between the different microcosm experiments differed somewhat in terms of their hygroscopicity, with the particles from the indoor MART being somewhat less hygroscopic ($GF(85\%) \sim 1.4 - 1.6$) than the particles from the outdoor MART ($GF(85\%) \sim 1.6 – 1.7$).

The observations here demonstrate that the climate impacts of marine-derived organic compounds can go beyond their demonstrated ability to influence cloud condensation nuclei efficacy (Quinn et al., 2014; Collins et al., 2013), additionally affecting the efficiency with which SSA particles scatter solar radiation. The implications of these results are explored here through calculations of the net decrease in the average per particle scattering that would theoretically result from substitution of less hygroscopic non-SS particles for SS-type particles (see Figure 6). This has been done for different assumed $GF_{non-SS}(85\%)$ values as a function of $f_{SS}$ using the average size distribution for the outdoor MART shown in Figure 2B. Given the particle size distributions measured here, this assessment pertains to submicron SSA, not the entire SSA particle size distribution observed over the ocean (which includes contributions from supermicron particles (Kleefeld et al., 2002)). For a given $GF_{non-SS}(85\%)$, the magnitude of the decrease varies linearly with $f_{SS}$. The range of $f_{SS}$ and $GF_{non-SS}(85\%)$ values determined here (about 0.4-0.7 and 1.4-1.7, respectively) correspond to decreases in scattering of about 10 to 35%. Thus, climate models that assume SSA particles behave like pure sea salt or NaCl (Stier et al., 2005; Schmidt et al., 2006) may over-predict SSA particle scattering, dependent upon the exact RH fields in the model. However, the range of $f_{SS}$ observed here, ~0.4 to 0.6, may be lower than in the ambient marine





atmosphere, given that the MART bloom experiments are more representative of regions of the ocean with high biological activity. For example, Lee et al. (2015) observed a slightly larger $f_{SS} \sim$ 0.72 in Bodega Bay, CA during a clean atmospheric period. Recent climate modeling studies (Partanen et al., 2014; O'Dowd et al., 2008) have attempted to account for variability in OM

fractions of SSA particles by parameterizing OM fraction as a function of Chl-a. However, relating the OM fraction of SSA particles to simple ocean biological metrics like Chl-a still remains challenging, as these metrics are often insufficient predictors for SSA particle composition (Quinn et al., 2014; Wang et al., 2015), and the measurements reported here indicate a clear lag between the peak in Chl-a and the minimum in the $GF(85\%)$ values. Quantitative understanding of the

climate impacts of SSA particles will require further understanding of the timing and relationships between ocean biogeochemistry and SSA properties.

## 5   Acknowledgments

This study was funded by the Center for Aerosol Impacts on Climate and Environment (CAICE), a NSF Center for Chemical Innovation (CHE-1305427). The authors thank all IMPACTS

participants and the SIO hydraulics facility staff. SDF and CDC additionally thank the students in ECI 247L during spring quarter 2015 at UC Davis, who validated the SEMS sizing.

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





**Table 1.** Summary of all instrumentation used in this study.

| Sampling Group | Group MART Flow Rates (LPM) Input/Output | Group Sampling Duration (hours) | Group Sampling Time After Group #1 (hrs) | Instrument/Method | Property Measured | Reference |
|---|---|---|---|---|---|---|
| *Particle Measurements* | | | | | | |
| 1 | 4.7 /3.7 | 1.5 | 0 | UCD Cavity Ringdown Spectrometer (CRD) | Light extinction by dry (<20% RH) and humidified (RH ~85%) particles | (Langridge et al., 2011; Cappa et al., 2012) |
| 1 | 4.7 /3.7 | 1.5 | 0 | Scanning Electrical Mobility Analyzer (SEMS) | Dry particle mobility size distributions (15–1000 nm) | |
| 1 | 4.7/3.7 | 1.5 | 0 | High Resolution Time of Flight Aerosol Mass Spectrometer (HR-ToF-AMS) | Bulk concentrations of non-refractory particulate components | (Canagaratna et al., 2007) |
| 2 | 3.9/2.9 | 2.0 | 9 | Aerosol Time of Flight Mass Spectrometer (ATOFMS) | Composition and number concentration of individual particles from 300 nm to 3000 nm | (Gard et al., 1998; Pratt et al., 2009) |
| 3 | 6.3/5.3 | 1.0 | 4.5 | Aerodynamic Particle Sizer (APS) | Dry particle aerodynamic size distributions (0.7-20 μm) | |
| *Waterside Measurements* | | | | | | |
| | | | | Aquaflour handheld portable fluorimeter | Chlorophyll-a | |
| | | | | High temperature combustion | Dissolved organic carbon | |



**Table 2.** Seawater conditions at the time of collection

| Date | Time | Chlorophyll-a (ug/L) | Water Temp. (°C) | Pressure (dbar) | Salinity (PSU) |
|------|------|----------------------|------------------|-----------------|----------------|
| 8 July | 12:00 | 0.998 | 23.1214 | 3.389 | 33.546 |
| 19 July | 12:00 | 2.171 | 20.7463 | 3.567 | 33.6051 |




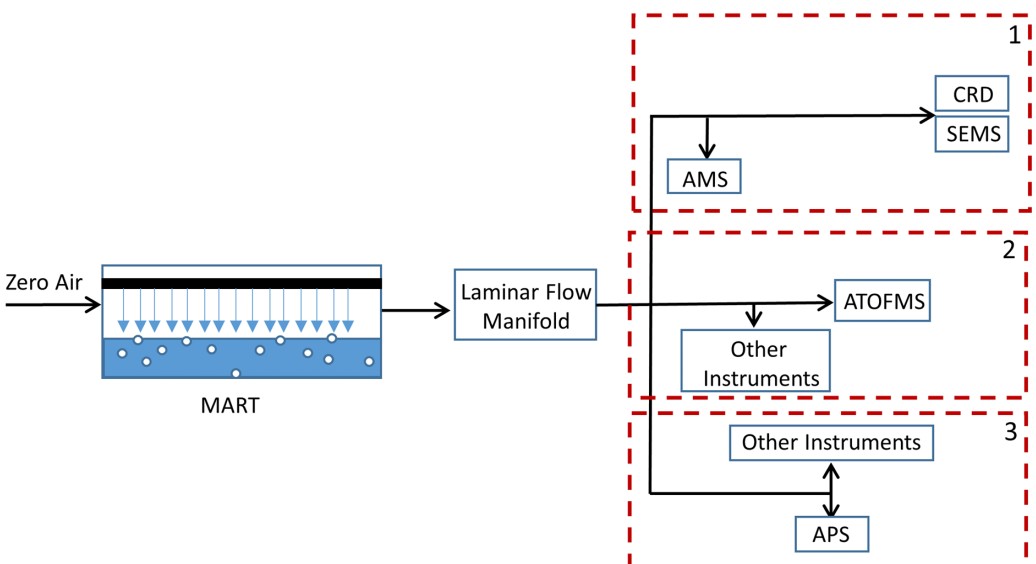

**Figure 1.** Experimental schematic for MART sampling during the IMPACTS 2014 study, with boxes labeled 1, 2, and 3 corresponding to different sampling configurations.



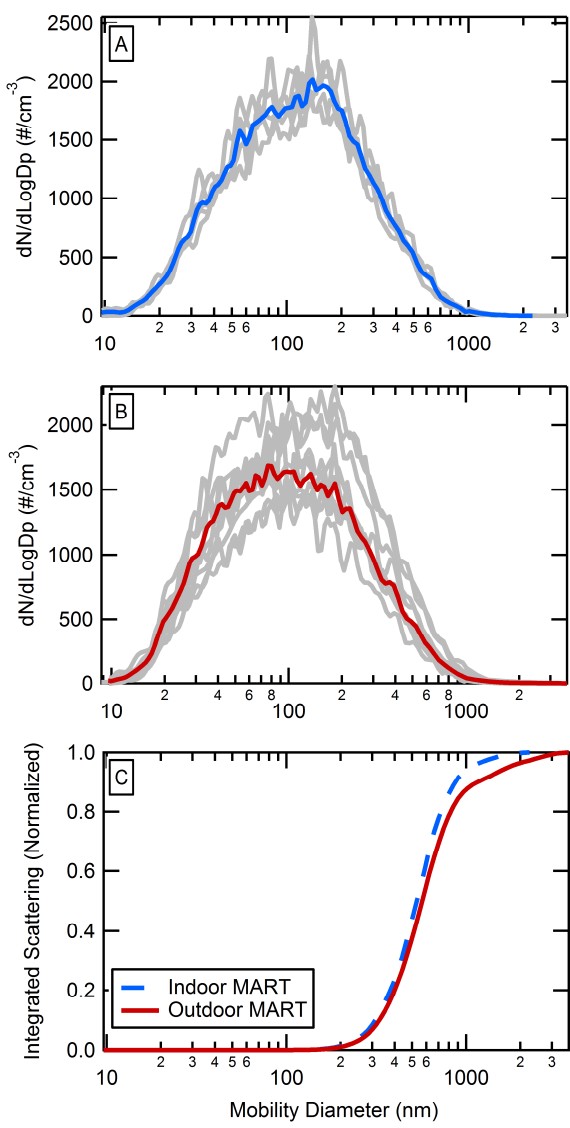

**Figure 2.** Single (grey) and average (red or blue) number-weighted merged size distributions for the **(A)** "indoor" and **(B)** "outdoor" MARTs averaged over the MART sampling period (1.5 hours) and **(C)** normalized integrated scattering as a function of dry mobility diameter for the merged size distribution. The optically-weighted median diameters are 530 nm for the indoor MART and 570 nm for the outdoor MART.





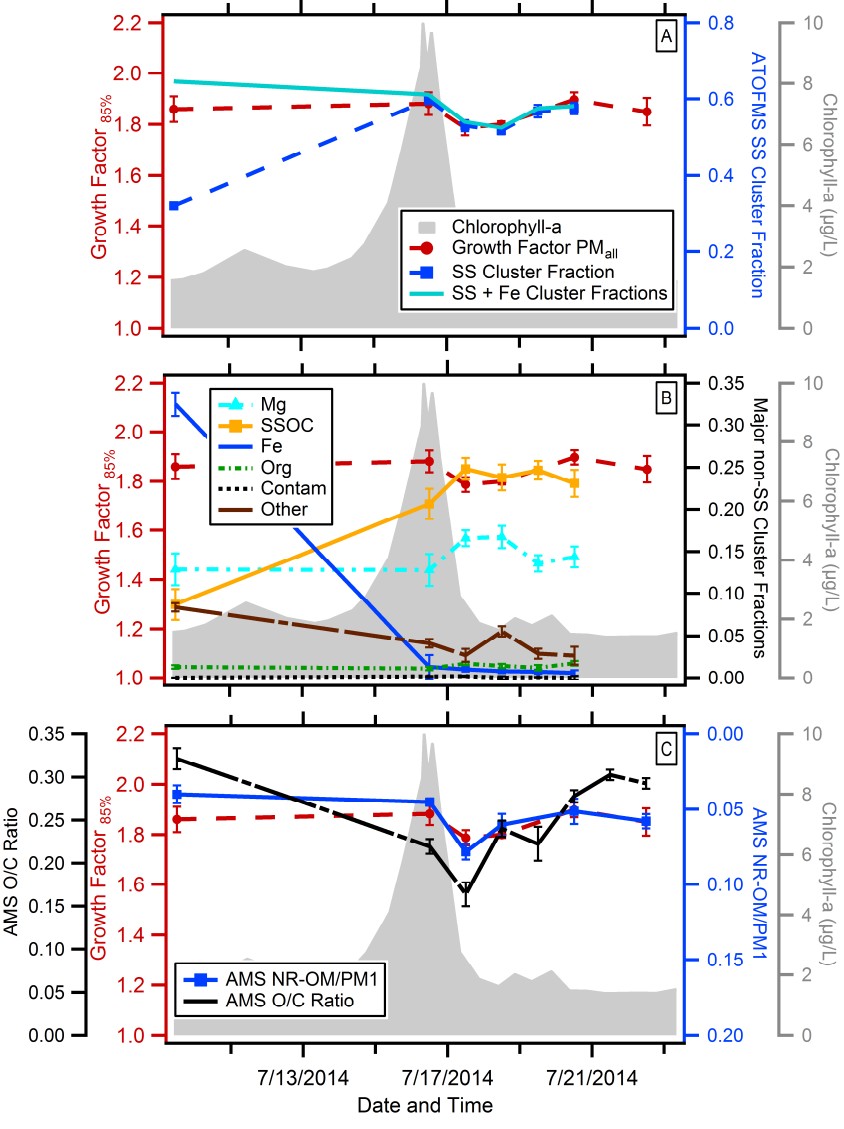

**Figure 3.** Time series for the indoor MART chlorophyll-a (gray), $PM_{all}$ $GF(85\%)$ (red circles), and **(A)** ATOFMS sea salt (SS) cluster fractions (blue dashed line) and ATOFMS SS + Iron type (Fe) cluster fractions **(B)** dominant non-sea salt cluster fractions magnesium (Mg) type (dashed turquoise line), Fe, "Other" type, and contamination (black line) and sea salt with organic carbon (SSOC) (orange line) cluster fractions, and **(C)** ratio of non-refractory organic matter (NR-POM)/ $PM_1$ mass (solid blue line) and the AMS O/C ratio (dashed black line). Note that the axis for NR-POM/$PM_1$ is reversed to facilitate comparison to $GF(85\%)$ values. The reported uncertainties for all properties is 1σ of the individual measurements over each sampling period.





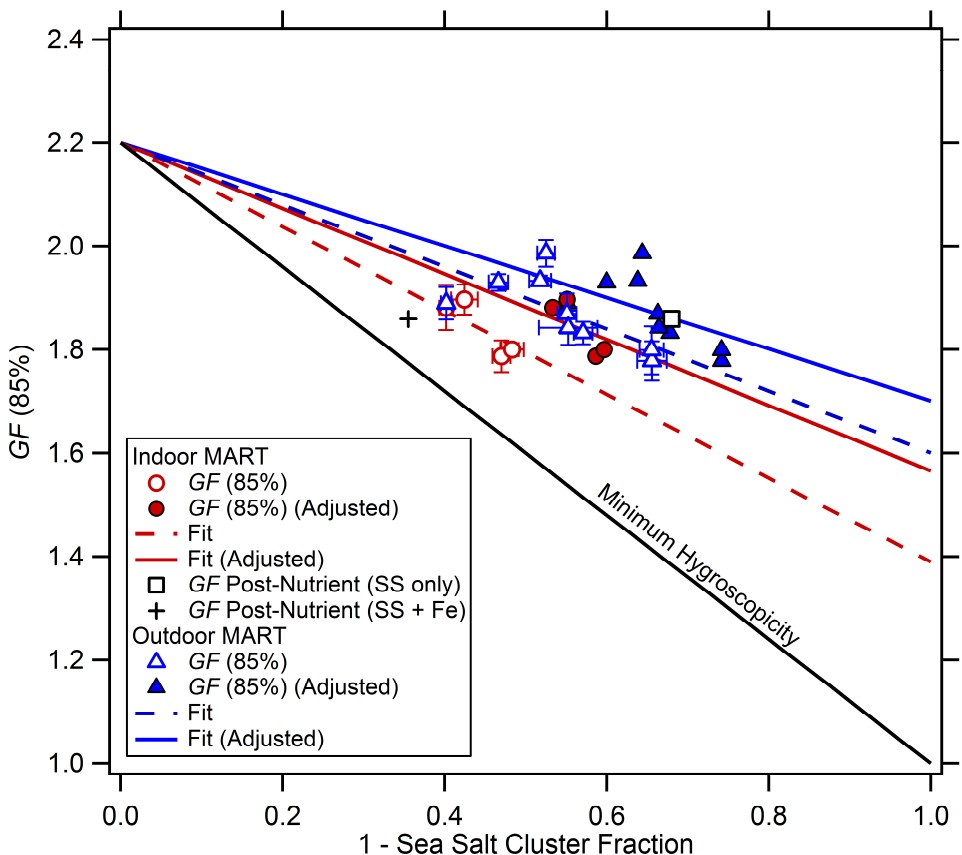

**Figure 4.** $PM_{all}$ $GF(85\%)$ as a function of the fraction of non-SS particles (equivalent to $1 - f_{SS,avg}$) for the indoor (open red circles) and the outdoor (open blue triangles) MARTs. Also shown as solid symbols are the $GF(85\%)$ values as a function of $1 - f_{SS*}$, where $f_{SS*}$ is the size-adjusted SS-type particle fraction (see main text for details). Fits to the data using Equation 5 are shown, assuming $GF_{SS}(85\%) = 2.2$ for the indoor overall (red dashed line), outdoor overall (blue dashed line), indoor adjusted (red solid line), and outdoor adjusted (blue solid line) MARTs. The overall retrieved $GF_{non-SS}(85\%)$ values were $1.39 \pm 0.03$ and $1.60 \pm 0.03$ for the indoor and outdoor MARTs, respectively. The adjusted $f_{SS,0.75}$ μm retrieved $GF_{non-SS}(85\%)$ values were $1.57 \pm 0.03$ and $1.70 \pm 0.03$ for the indoor and outdoor MARTs, respectively. The black solid line connecting $GF_{SS}(85\%) = 2.2$ and $GF_{non-SS}(85\%) = 1.0$ provides the minimum value expected for any combination of SS and non-SS particles. The post-nutrient $GF(85\%)$ corresponding to $f_{SS}$ (open black square) and $f_{SS+Fe}$ (black cross) are shown for reference, but not included in the fits.





**Figure 5.** Same as Figure 3 above, but for the outdoor MART.




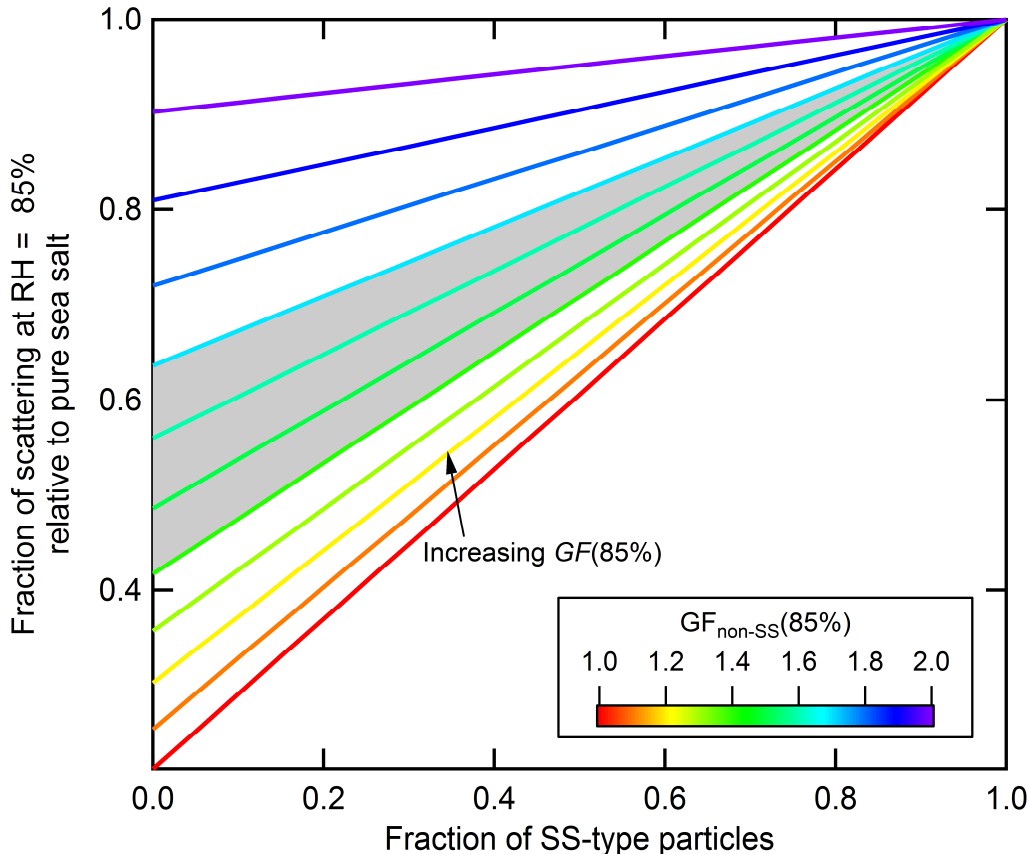

**Figure 6**. Calculated fraction of scattering relative to pure sea salt particles at 85% RH as a function of the number fraction of SS-type particles, assuming all non-SS particle types have the same, constant hygroscopicity and a refractive index of 1.55. The different curves are colored according to the assumed $GF_{non-SS}(85\%)$ value, ranging from 1.0 to 2.0, given $GF_{SS}(85\%) = 2.2$. The gray band shows the range of $GF_{non-SS}$ values indicated by the current measurements.