# Peer review of "Linking variations in sea spray aerosol particle hygroscopicity to composition during two microcosm experiments"

_Atmospheric Chemistry and Physics, 2016_

## Referee Comment (RC1) · Anonymous Referee #1 · 2 Mar 2016

General comment:

The paper by Forestieri et al. presents results from two microcosm experiments on the properties of sea spray aerosols, focusing on their hygroscopic and optical properties, as a function of the seawater composition. The seawater composition was artificially modified in the microcosms by the addition of nutrients. The authors infer an average hygroscopic growth factor (HGF) for the whole sea spray aerosol population, from the measurement of the aerosol extinction enhancement due to the uptake of water vapour at 85% humidity. Results show a decrease of the HGF by 10 to 19% relative to pure inorganic sea salt. The authors then infer an average chemical composition from the HGF with the hypothesis that the organic fraction is hydrophobic. No linear

link between the increase of Chl-a levels and the change in aerosol chemical composition (organic content, mainly) was observed. The study of the impact of the presence of organic matter in primary sea spray on its optical properties through the effect of a decreased water uptake has never been investigated in the past to my knowledge. Whether this has an important impact or not is important. In this view this is a very valuable study. However, the measurement methodology relay on several hypothesis and approximations that could be better justified (see detailed comments), and the article is more focusing on inferring the organic fraction of primary organic aerosol than on evaluating this impact, which could be more emphasised (the impact on scattering is only mentioned in the conclusion as a range from 10 to 35% for 85% humidities). I would have expected a time series of the extinction (wet and dry) in order to directly evaluate the impact of a phytoplanktonic bloom on the optical properties of sea salt aerosol. I recommend publication after major revisions.

Detailed comments

Page 4, lines 6-7 : "SSA particles sampled from the MARTs are primary, since the average residence time in the MARTs is much shorter than the time scale required for secondary processing of SSA particles (e.g. heterogeneous gas-phase reactions) (Lee et al., 2015)." What is the residence time in the microcosm headspace, what are briefly the results from Lee et al. 2015 to support this hypothesis ? How can the absence of any photochemical reactions producing condensing organic matter be excluded?

Page 5 lines 13_18 : "The same seawater as used in the indoor MART was added to a separate MART and sampled immediately after collection and before nutrient addition. However, the resulting particle size distribution from this MART differed substantially from those measured from the indoor MART, with a much greater contribution of large particles. Thus, the measurements from this separate MART are not directly comparable to the measurements from the indoor MART and are not considered further" Is there any explanation for this ? Could it be that the same difference in original size distribution (before enrichment) was observed in the outdoor experiment ?

[Figure]

Page 6, line 10 : "Group 1 sampled for 1.5 h, group 2 sampled for 2 h, and Group 3 for sampled 1 h each day that sampling was conducted" Was sampling always performed in this order ? Can there be a bias due to the position of the sampling period during the day ? Has this been tested ?

Page 6, line 25 : "The SEMS and APS distributions were merged using the SEMS distribution up to 1 $\mu$m and the (dp,m equivalent) APS distribution at larger diameters." How did the two instrument compare on their common size range ? Why was the APS preferred over the APS on the 1-1.9 micron size range ?

Page 7, line 2 : "Light absorption by the SSA particles was negligible, and thus extinction is equal to scattering, i.e. bext = bsca." Was this assumption validated ? Can Brown carbon contribute the SSA absorption ?

Page 9, lines 16-17 : "Unlike f(RH), GF values are independent of the dry particle size, and thus only depend on composition" This is only true for larger particles, the smaller the particles the highest the kelvin effect is. Maybe it is useful to argue that this hypothesis is true for the sizes of particles relevant here.

Pages 9 and 10 : calculation of GF(RH) : the underlying hypothesis for such a iterative calculation is that the chemical composition of the aerosol is homogeneous over the whole range of sizes (independent of the particle diameter). Figure S6 does not show this. How does this impact the results ?

Page 10, lines 1ç-21 : "The measured GF(85%) for NaCl was 2.09 +/- 0.03 and for (NH4)2SO4 was 1.59 +/- 0.05, which compare very well with literature values of ∼2.1 for NaCl (Cruz and Pandis, 2000; Laskina et al., 2015; Hansson et al., 1998) and ∼1.55 for ammonium sulfate (Laskina et al., 2015; Wise et al., 2003)." The literature values should be reported for a given aerosol size (or size range).

Page 12 , lines 13_14 : "Uncertainty in the assumed RI value for the dry particles may explain a small fraction (<5%) of the difference." How was this assessed ? Has

the chemical analysis of the aerosol been used to estimate the real RI? All hypothesis for possible discrepancies addressed in this paragraph should be detailed in the methodology section (or at least in the supplementary material). An overall uncertainty on the HGF retrievals procedure should be calculated and compared to the measured HGF variability and consequent Org frac variability, so the reader can be convinced that the measured time variations are real. The uncertainty on the calculation method should be less than the 10 to 19% decrease in HGF for the results of the paper to be significant.

Page 12, lines 21-24 : "We have tested the sensitivity of the retrieval method to an 8% increase in the particle diameters. The retrieved GF values are increased by a marginal amount (0.015-0.03) when the diameters are increased, and thus such potential sizing uncertainty does not affect the main conclusions presented here" Does this mean that a particle diameter increase of 8% was actually applied to the data set ?

Page 17, lines 10-12 : "By using a campaign-average scaling factor, it is implicitly assumed that the actual variations in $f_{SS,0.75\mu m}$ are captured by $f_{SS,avg}$, which seems reasonable given the general constancy of the size distributions over the course of each of the microcosm experiments, c.f. Fig. 2." Why would the relative stability of the size distribution shown on fig 2 insure that the non-sea salt content of the aerosol (shown to increase in the course of the experiment) evolves uniformly with size ?

Page 17, lines 19-24 : "...organic volume fraction ($\varepsilon i$) of 0.56 – 0.88 for these particle types if it is assumed that volume mixing rules apply (i.e. the Zdanovskii-Stokes-Robinson mixing rules (Stokes and Robinson, 1966)). Since the non-SS values range from 0.53 – 0.74, if it is assumed that the SS-type and non-SS particle types have similar size distributions, then the implied ensemble average $\varepsilon i$ would be about 0.33 – 0.52." I understand that the first time that $\varepsilon i$ is used it refers to the fraction of hydrophobic material in the non-SS fraction, while the second time it is used it refers to the fraction of hydrophobic material in the overall aerosol. If this is right, the same terminology should not be used for both.

Page 18, lines 9-12: "Both DOC and heterotrophic bacteria concentrations increased as the bloom progressed until they stabilized around the point when Chl-a concentrations had returned approximately to their pre-bloom levels, with DOC concentrations ranging from 200 to 300 $\mu$M C and heterotrophic bacteria concentrations from 1 x 106 to a peak of 1.7 x 107 mL-1 (Figure S5B)." Are those values realistic for natural seawaters ?

Technical comments

Page 5, line 8 : mesocosm or microcosm ?

Figure 3 (B) : description of Org not in the figure text. "the reported uncertainties for all properties is 1 sigma. . . "should be "the reported standard deviations for all properties is 1 sigma. . ." as those are not uncertainties on the measurements

---

## Referee Comment (RC2) · Anonymous Referee #2 · 22 Mar 2016

The paper by Forestieri et al. reports on hygroscopicity of sea spray particles generated in lab conditions during various stages of phytoplankton bloom development. Lab generated sea spray studies are being pursued by many research groups during recent years trying to uncover the mechanisms and impacts of organic matter enrichment in sea spray particles.

The hygroscopic properties of sea spray were studied by measuring scattering properties of wet versus dry particles. As it measures bulk sea spray population it is missing on the important aspect of size dependent chemical composition which is critical in uncovering organic matter enrichment processes. The results of the study are not particularly new and the authors could increase its significance by assessing radiative

forcing impacts. It would be very interesting how the results of this study compare with the study by Vaishya et al. (2013) conducted in marine atmosphere (the study referenced, but not discussed).

The most confusing aspect of this study is that a significant change in hygroscopicity of sea spray particles is only loosely connected to chemical composition. AMS did not detect the amount of organic matter required to explaining the observed change in GF. While the authors speculate about the bounce and refractory nature of sea spray particles (providing no references) the published evidence is in favour of AMS being able to quantitatively measure sea spray e.g. (Allan et al., 2004; Ovadnevaite et al., 2012; Schmale et al., 2013) to mention a few. ATOFMS results seem to correlate with the observed GF, but ATOFMS lacks quantitative estimate as its sensitivity to sea spray is rather poor. As the mixed-in organic matter in sea spray would increase ATOFMS sensitivity, the amount of non-sea-salt particles would be biased high. Also considering ATOFMS size range and MART sea spray particle size peaking at a size where ATOFMS just starting to detect particles, it appears that ATOFMS measured only a fraction of sea spray population. As it currently stands, the data do not corroborate each other.

Other comments

Page4, Line 24. I wonder if the flow was split isokinetically (equal face velocities) between instruments sampling from MART as that could affect sampled particle sizes of individual instruments. The authors mentioned laminar conditions, but laminar conditions limit particle losses to tubing walls while isokinetic split maintains the same particle population into each sampling line.

Page 5, Line 12. Peak chlorophyll concentration was mentioned as 10ug/l in the previous paragraph.

Line 17. Was this MART reproducibility issue or else?

Line 26. Considering 3week duration of the whole experiment a substantial degradation of organic matter (rotting) should have occurred at ambient temperatures in excess of 25C. Was bacteria growth monitored to inform on such process and if not informative, how could that be related to real world environment?

Page 6, Line 27. Were the particles dried? What RH? It seems that APS density was picked based on OM fractional contribution which suggests about 30% depending on OM density. If particles were not dried the picked density would not apply.

Page 7, Line 26. Was PM2.5 cyclone operated in dry or wet conditions which could have converted PM2.5 into PM1 or lower size cut if wet?

Page 8, Line 3. Following the paragraph above referring to minimal contribution of >2.5um particles to the total SSA population it follows that ATOFMS sampled minor fraction of particles considering its transmission efficiency. Given low ATOFMS sensitivity to sea salt particles it transpires that ATOFMS sampled fraction of a fraction of SSA population. This aspect has to be clearly articulated otherwise references to SSA chemical composition is heavily biased towards supermicron particles.

Page 9, Line 2. Is it referred to dry of wet particles? If SEMS was dried, but AMS was not then not same SSA population was measured by the two instruments making diameter match irrelevant. Wet particles entering the AMS inlet are instantly frozen due to adiabatic expansion and segregated by aerodynamic lenses based on their wet diameter. Assuming RH in the MART and subsequent sampling lines 90-100%, wet particle diameter was 2-3 times larger than dry SEMS particles. NR-OM mass was therefore limited to 186-280nm instead of 560nm. The drying issue appears quite central throughout the manuscript, so I suggest it clarifying at the beginning and using notations d(dry), d(wet) were appropriate. If AMS sampled wet particles that would explain the missing mass discussed few lines below.

Line 8. AMS is typically calibrated with dry NH4NO3 particles. Why would SS particles bounce more than the calibration particles as AMS chemical species mass is calculated

on nitrate equivalent basis?

Page 11, Line 20. Many lab and ambient studies reported chemical composition dependence on particle size which would make GF size dependent too. This study reports size independent (averaged) GF which is rather misleading and, therefore, the issue should be clearly stated.

Line 28. The discrepancy can be partly due to shallow cut-off function of PM2.5 cyclon. Another source of discrepancy can be due to losses of wet particles and corresponding losses in dryers as in general wet particles are lossier. Again the drying of the particle is very unclear throughout the study and difficult to interpret.

Page 12, Line 6. Wiedensohler et al. (2012) reported that in general sizing errors of different instruments can be objectively up to 10%.

Page 13, Line 13. 2.2 at 85% or 90%? Also on page 10, GF(85%) of NaCl was referred to as 2.1.

Line 27. Is it possible that the relative abundance of Fe-rich particles was due to higher sensitivity of ATOFMS to Fe-rich versus SSA?

Page 14, Line 23. This is only true if ATOFMS and CRD size ranges were exactly the same which was not the case as ATOFMS cannot reliably detect 100nm particles, especially SSA.

Page 16, Line 13. Page 10 referred to 2.1 GF(85%). Why GF=1 is expected as the minimum combined value? Any reference to backup? Marine gels and micelles have been reported to process some water despite being generally hydrophobic (Ellison et al., 1999; Chakraborty and Zachariah, 2007). Fatty acid is only one of the many possible compounds and necessarily entirely hydrophobic.

Line 20. It has been demonstrated in numerous studies that OM fraction in sea spray is size dependent. Should the GF value of 1.39 be interpreted as a bulk average of highly enriched and poorly enriched SS particles?

Page 17, Line 20. There is an issue regarding size dependent chemical composition. As scattering is dominated by larger submicron sizes and the smaller submicron particles tend to be more enriched in OM, averaged GF of this study missing out on the important aspect of size dependent chemical composition.

Line 24. How this volume fraction compared with AMS chemical composition? Did AMS record any substantial organics as 0.33-0.52 volume fraction would suggest? Figures show that AMS OM fraction was 0.05.

Page 18, Line 5. "which was 5 times higher". Much higher chl was probably due to higher temperature than the ocean (what was the T range?) and plentiful nutrients.

Page 19, Line 3. Consider different size ranges sampled if AMS was not dried.

Table 1. AMS size range is missing.

Allan, J. D., Bower, K. N., Coe, H., Boudries, H., Jayne, J. T., Canagaratna, M. R., Millet, D. B., Goldstein, A. H., Quinn, P. K., Weber, R. J., and Worsnop, D. R.: Submicron aerosol composition at Trinidad Head, California, during ITCT 2K2: Its relationship with gas phase volatile organic carbon and assessment of instrument performance, J. Geophys. Res.-Atmos., 109, 10.1029/2003jd004208, 2004. Chakraborty, P., and Zachariah, M. R.: "Effective" negative surface tension: A property of coated nanoaerosols relevant to the atmosphere, Journal of Physical Chemistry A, 111, 5459-5464, 10.1021/jp070226p, 2007. Ellison, G. B., Tuck, A. F., and Vaida, V.: Atmospheric processing of organic aerosols, Journal of Geophysical Research: Atmospheres, 104, 11633-11641, 10.1029/1999JD900073, 1999. Ovadnevaite, J., Ceburnis, D., Canagaratna, M., Berresheim, H., Bialek, J., Martucci, G., Worsnop, D. R., and O'Dowd, C.: On the effect of wind speed on submicron sea salt mass concentrations and source fluxes, J. Geophys. Res.-Atmos., 117, 10.1029/2011jd017379, 2012. Schmale, J., Schneider, J., Nemitz, E., Tang, Y. S., Dragosits, U., Blackall, T. D., Trathan, P. N., Phillips, G. J., Sutton, M., and Braban, C. F.: Sub-Antarctic marine aerosol: dominant contributions from biogenic sources, Atmos. Chem. Phys., 13,

8669-8694, 10.5194/acp-13-8669-2013, 2013. Vaishya, A., Ovadnevaite, J., Bialek, J., Jennings, S. G., Ceburnis, D., and O'Dowd, C. D.: Bistable effect of organic enrichment on sea spray radiative properties, Geophys. Res. Lett., 40, 6395-6398, 10.1002/2013gl058452, 2013. Wiedensohler, A., Birmili, W., Nowak, A., Sonntag, A., Weinhold, K., Merkel, M., Wehner, B., Tuch, T., Pfeifer, S., Fiebig, M., Fjaraa, A. M., Asmi, E., Sellegri, K., Depuy, R., Venzac, H., Villani, P., Laj, P., Aalto, P., Ogren, J. A., Swietlicki, E., Williams, P., Roldin, P., Quincey, P., Huglin, C., Fierz-Schmidhauser, R., Gysel, M., Weingartner, E., Riccobono, F., Santos, S., Gruning, C., Faloon, K., Beddows, D., Harrison, R. M., Monahan, C., Jennings, S. G., O'Dowd, C. D., Marinoni, A., Horn, H. G., Keck, L., Jiang, J., Scheckman, J., McMurry, P. H., Deng, Z., Zhao, C. S., Moerman, M., Henzing, B., de Leeuw, G., Loschau, G., and Bastian, S.: Mobility particle size spectrometers: harmonization of technical standards and data structure to facilitate high quality long-term observations of atmospheric particle number size distributions, Atmospheric Measurement Techniques, 5, 657-685, 10.5194/amt-5-657-2012, 2012.

---

## Author Comment (AC1) · 2 Jun 2016

**Author's response to reviewer's comments**

We thank the two reviewers for their constructive comments and suggestions, which have helped to improve the manuscript. Before responding to the specific individual comments from the reviewers, we note that, we have made substantial changes to the manuscript based on the reviewer comments. Specifically, we changed the focus from looking at the relationship between sea spray aerosol particle hygroscopicity and ATOFMS cluster-type fractions to one between hygroscopicity and organic matter volume fractions ($\varepsilon_{org}$). The OM volume fractions were estimated from the AMS organic matter/$PM_1$ mass fractions that were presented in the original manuscript. In the original manuscript, we did not use the $\varepsilon_{org}$ quantitatively, as there are concerns regarding the detection efficiency of the AMS for these marine derived organics as particles containing a large fraction of sea salt have a higher susceptibility to particle bounce and organic matter contained in these particles may be inefficiently vaporized (as is suggested by the results presented by Frossard et al. (2014)). That said, in one of the references mentioned by Reviewer #2 (Ovadnevaite et al. (2012)), it was determined that sea salt aerosol had a collection efficiency ($CE$) in the AMS of 0.25. We have therefore now corrected the AMS organic matter/$PM_1$ mass fractions using a $CE$ of 0.25, and the mass fractions were converted to $\varepsilon_{org}$ assuming a density of 1 $g/cm^3$ for organic matter. The resulting $\varepsilon_{org}$ are therefore relatively uncertain in terms of absolute magnitude, but the trends with time should be reasonably robust under the assumption that the $CE$ did not change substantially across the measurement campaign. A thorough discussion of the uncertainties in $\varepsilon_{org}$ as estimated from AMS organic matter/$PM_1$ mass fractions and the details for calculating $\varepsilon_{org}$ has been added to section 2.2.1.

> "It is important to note that while the temporal trends of the AMS NR-OM/$PM_1$ fractions are likely reflective of the general behavior, the absolute values are more difficult to quantify because NR-OM associated with particles containing high sea salt fractions may not be vaporized efficiently by the AMS due to the refractory nature of sea salt (Frossard et al., 2014) and to the susceptibility of SSA particles to particle "bounce" in the AMS. Consequently, the SSA particles, including the NR-OM component, are detected with a collection efficiency ($CE$) lower than unity (Frossard et al., 2014). One previous study (Ovadnevaite et al., 2012) determined the $CE$ value for organic-free sea salt sampled when RH < 70% is approximately 0.25. However, they also note that the $CE$ is potentially instrument dependent, and further may not be applicable to the organic fraction in sea spray particles due to differences in ionization efficiency (which is a component of the overall $CE$) (Ovadnevaite et al., 2012). It is also possible that the $CE$ differs between particles that have differing relative amounts of OM and sea salt. Despite such uncertainties in quantification of NR-OM by the AMS for sea spray particles, the NR-OM mass concentrations for the sampled SSA particles were determined in this study assuming $CE = 0.25$. The measured NR-OM mass concentrations were used to calculate NR-OM volume concentrations assuming a density ($\rho$) of 1.0 $g/cm^3$. A value of 1.0 $g/cm^3$ for $\rho_{OM}$ is consistent with that of fatty acids ($\rho < 1$ $g/cm^3$), which are a significant fraction of marine-derived OM (Mochida et al., 2002; Cochran et al., 2016). However, this value serves as a lower bound for $\rho_{OM}$ because OM with higher densities, such as sugars ($\rho \sim 1.7$ $g/cm^3$), have also been observed in SSA (Quinn et al., 2015). The NR-OM volume

fractions of SSA ($\varepsilon_{org}$) were calculated as the ratio between the observed NR-OM volume concentrations and the integrated total particle volume concentrations from the size distribution measurements. Given the use of a lower-limit value for $\rho_{OM}$ the $\varepsilon_{org}$ are likely upper limits (not accounting for uncertainty in the assumed $CE$)."

The CE-corrected $\varepsilon_{org}$ are now used as the primary compositional metric for understanding both the depression in $GF(85\%)$ values relative to inorganic sea salt and their temporal variability. Figures 3 and 5 have been updated to show the CE-corrected $\varepsilon_{org}$ values. Discussion regarding the temporal variability in and absolute magnitude of the CE-corrected $\varepsilon_{org}$ has been added.

"The NR-OM volume fractions of SSA varied from 0.29 to 0.50 throughout the course of the indoor MART microcosm experiment (Figure 3). The observation of such large $\varepsilon_{org}$ values is consistent with the substantial depressions in the $GF(85\%)$ values relative to pure, inorganic sea salt (2.1). The temporal variation in the $\varepsilon_{org}$ was generally similar to that of the $GF(85\%)$ values, with smaller $GF(85\%)$ values corresponding to larger $\varepsilon_{org}$ values, although the peak in $\varepsilon_{org}$ is somewhat sharper than the dip in the $GF(85\%)$. The inverse relationship between the $GF(85\%)$ and $\varepsilon_{org}$ is consistent with organic compounds being less hygroscopic than sea salt."

The original Figure 4, which showed the relationship between the $GF(85\%)$ values and the ATOFMS non-sea salt cluster fractions, has been replaced. The new Fig. 4 now shows the relationship between $GF(85\%)$ and the CE-corrected $\varepsilon_{org}$. We now use the Zdanovskii-Stokes-Robinson (ZSR) mixing rules to estimate a $GF(85\%)$ value for the organic matter component of the SSA particles specifically. The fitting procedure is described on Page 16 Lines 12-20 in the updated manuscript.

There has also been an evolution in our understanding of the ATOFMS clusters determined for nascent sea spray since the manuscript was originally submitted. A portion of the non-sea salt (sodium-depleted) clusters can be explained by the incomplete ionization of sea salt particles (Sultana et al., In Prep.). This change in our understanding of ATOFMS cluster types further supports our decision to use the CE-corrected $\varepsilon_{org}$ from the AMS in place of ATOFMS cluster types to understand the dependence of the $GF(85\%)$ on variations in particle composition. A brief discussion of this new understanding regarding the ATOFMS clusters for nascent sea spray has been added to the manuscript in the methods section.

"It is important to note, however, that dried SSA particles sampled by the ATOFMS can be spatially chemically heterogeneous, with shells depleted in Na and rich in Mg, K, and Ca (Ault et al., 2013). Thus, some fraction of the particles identified as having Mg or SSOC type spectra may be partially explained by the incomplete ionization of sea salt particles (Sultana et al., In Prep.). However, variations in the thickness of this Na-depleted shell likely reflect variations in the total particle organic content. Therefore, increases in the fraction of SSOC or Mg type mass spectra generated suggest a net increase in SSA particle organic content."

We made other changes, where deemed appropriate and added additional figures and tables as supplemental materials. Our point-to-point response to both reviewers follows below.

**Key: Black = Reviewer, Blue = Response**

**Response to Reviewer #1**

The paper by Forestieri et al. presents results from two microcosm experiments on the properties of sea spray aerosols, focusing on their hygroscopic and optical properties, as a function of the seawater composition. The seawater composition was artificially modified in the microcosms by the addition of nutrients. The authors infer an average hygroscopic growth factor (HGF) for the whole sea spray aerosol population, from the measurement of the aerosol extinction enhancement due to the uptake of water vapour at 85% humidity. Results show a decrease of the HGF by 10 to 19% relative to pure inorganic sea salt. The authors then infer an average chemical composition from the HGF with the hypothesis that the organic fraction is hydrophobic. No linear link between the increase of Chl-a levels and the change in aerosol chemical composition (organic content, mainly) was observed. The study of the impact of the presence of organic matter in primary sea spray on its optical properties through the effect of a decreased water uptake has never been investigated in the past to my knowledge. Whether this has an important impact or not is important. In this view this is a very valuable study. However, the measurement methodology relay on several hypothesis and approximations that could be better justified (see detailed comments), and the article is more focusing on inferring the organic fraction of primary organic aerosol than on evaluating this impact, which could be more emphasised (the impact on scattering is only mentioned in the conclusion as a range from 10 to 35% for 85% humidities). I would have expected a time series of the extinction (wet and dry) in order to directly evaluate the impact of a phytoplanktonic bloom on the optical properties of sea salt aerosol. I recommend publication after major revisions.

Regarding the reviewers comment as to presentation of a time-series of extinction (wet and dry), we note that for this study the absolute values of extinction are not nearly as important as the relative values between the humidified and dried extinction (i.e. f(RH) = wet extinction/dry extinction). Although MARTs produce particles with size distributions that are similar to particles produced from breaking waves in the ocean, the absolute particle concentrations (and therefore the absolute scattering) is different than what would be observed in the ambient marine atmosphere, as the absolute values depend on the sample flow rates, MART size, plunging frequency, etc. (Stokes et al., 2013). Thus, a focus on the extensive properties (e.g. dry and wet scattering) is, in our opinion, not as important as a focus on the intensive properties (e.g. f(RH)). In the original manuscript, we presented a time-series only growth factor values that were derived from the f(RH) measurements and the size distribution measurements, not the f(RH) measurements themselves. However, to address the reviewers concern, we have now added to the supplemental material a time-series of measured wet and dry extinction measurements and of the associated f(RH).

Regarding discussion of the limited evaluation of the impact of organic material on scattering, we intentionally kept this discussion "simple". The reason for this is that the ultimate impacts depend on not only variability in the organic fraction, but also in real variations in relative humidity fields. Thus, a comprehensive evaluation would likely require assessment within a climate model, which is outside the scope of this work. To address the reviewers suggest that we

emphasize the potential impact to a greater extent, we have added the following sentences to the conclusions:

> "Regardless, the results presented here suggest that OM in SSA particles may have a non-negligible, yet variable impact on the light scattering by SSA particles in the ambient atmosphere. Most likely, the simulated cooling effect of SSA particles due to aerosol-radiation interactions (i.e. the "direct effect") would be decreased relative to the assumption that all SSA behaves as sea salt."

Therefore, f(RH) (= wet scattering/dry scattering), which is an intensive parameter, is used to assess radiative impacts. A figure comparing the f(RH) of particles observed in this study to the f(RH) expected for pure, inorganic sea salt calculated using average size distributions for both microcosm experiments has been added to the supplementary material (Figure S8).

Page 4, lines 6-7 : "SSA particles sampled from the MARTs are primary, since the average residence time in the MARTs is much shorter than the time scale required for secondary processing of SSA particles (e.g. heterogeneous gas-phase reactions) (Lee et al., 2015)." What is the residence time in the microcosm headspace, what are briefly the results from Lee et al. 2015 to support this hypothesis ? How can the absence of any photochemical reactions producing condensing organic matter be excluded?

Oxidants which accompany secondary processes, such sulfates and nitrates, were not present in the ATOFMS spectra. This is as expected as zero air (particle, ozone, and volatile organic species free) was used to feed the MART headspace, so secondary processes should be highly minimized even over extended periods. Hydroxyl radicals should not be generated by the lamps used here, as the higher-energy UV radiation needed to photolyze water would be filtered by the acrylic chamber walls. The residence time in the headspace is particle size dependent (as discussed in Stokes et al., 2013). At the flow rate used here, for particles in the size range 400-600 nm the e-folding lifetime was 11 minutes and in the size range 1-2 microns was 8.5 minutes.

Page 5 lines 13_18 : "The same seawater as used in the indoor MART was added to a separate MART and sampled immediately after collection and before nutrient addition. However, the resulting particle size distribution from this MART differed substantially from those measured from the indoor MART, with a much greater contribution of large particles. Thus, the measurements from this separate MART are not directly comparable to the measurements from the indoor MART and are not considered further" Is there any explanation for this ? Could it be that the same difference in original size distribution (before enrichment) was observed in the outdoor experiment ?

The large supermicron mode observed during in the pre-nutrient period may be due to differences in MART conditions, such as water level or sampling tube length, but we cannot entirely rule out that it is some difference due to the nutrient addition. Unlike the indoor MART, the outdoor MART was not sampled prior to enrichment, and thus we cannot directly address the reviewers' second question. In separate (unpublished) MART experiments, strong differences in the size distributions before/after nutrient addition are not observed, suggesting that in this particular case the size distribution difference was driven by external factors (which resulted

from the somewhat more complex experimental setup during the IMPACTS study due to the large number of instruments involved).

Page 6, line 10 : "Group 1 sampled for 1.5 h, group 2 sampled for 2 h, and Group 3 for sampled 1 h each day that sampling was conducted" Was sampling always performed in this order ? Can there be a bias due to the position of the sampling period during the day ? Has this been tested ?

On all days, except for 7/9, the instruments sampled at the same time each day. It is possible that the composition differed when the cavity-ringdown (CRD) sampled versus when the ATOFMS sampled, but this has not been systematically tested. In fact, some differences are to be expected given that the observations suggest changes in composition and f(RH) from one day to the next. However, the correlation between the ATOFMS organic markers and the CRD growth factors indicates that compositional changes were gradual and the composition of particles sampled by both instruments was consistent on a given day, even though the ATOFMS sampled 9 hours after the CRD.

Page 6, line 25 : "The SEMS and APS distributions were merged using the SEMS distribution up to 1 μm and the (dp,m equivalent) APS distribution at larger diameters." How did the two instrument compare on their common size range ? Why was the APS preferred over the APS on the 1-1.9 micron size range ?

A comparison between the SEMS and APS suggests that the SEMS under-counted particles with mobility diameters > 1 micron. This was characterized during separate experiments in which substantial fractions of supermicron particles were sampled. At just above 1100 nm, the SEMS undercounted by ~20% and at 1.5 μm, the SEMS undercounted by ~90%. We suspect that this difference resulted from the SEMS not being optimized in these experiments to transmit larger particles, and thus internal losses increased as size increased. It is for this reason that we chose to use the merged distribution, as opposed to the SEMS distribution by itself. However, in the size region that is most relevant to the experiments considered here, namely below ~ 1 micron, there was no substantial difference between the SEMS and merged SEMS + APS distributions. Since the fraction of supermicron particles was very small for these MART studies the impact of SEMS vs. APS differences at larger sizes is negligible.

Page 7, line 2 : "Light absorption by the SSA particles was negligible, and thus extinction is equal to scattering, i.e. bext = bsca." Was this assumption validated ? Can Brown carbon contribute the SSA absorption ?

The SSA absorption was measured by a UC Davis photoacoustic spectrometer (Lack et al., 2011; Cappa et al., 2012) and the observed absorption was 0 $Mm^{-1}$ (within uncertainty), whereas the measured extinction was ~250 $Mm^{-1}$.

Page 9, lines 16-17: "Unlike f(RH), GF values are independent of the dry particle size, and thus only depend on composition" This is only true for larger particles, the smaller the particles the highest the kelvin effect is. Maybe it is useful to argue that this hypothesis is true for the sizes of particles relevant here.

The reviewer raises an important point. We have therefore calculated theoretical growth factors as a function of particle diameter using equation 11 from Petters and Kreidenweis (2007), assuming 85% and assuming $\kappa = 1.3$. The relevant equation is:

$$\frac{RH}{\exp\left(\dfrac{A}{D_{dry}} \cdot GF\right)} = \frac{GF^3 - 1}{GF^3 - (1 - \kappa)}$$

The Kelvin effect is inherent in the $A$ term in the above equation since:

$$A = \frac{4 \cdot \sigma \cdot MW_{H2O}}{RT \cdot \rho_{H2O}}$$

where $\sigma$ = surface tension, $MW_{H2O}$ is the molecular weight of water and $\rho_{H2O}$ is the density of water. Results of these calculations are shown in the figure below. The black line indicates where the median (50%) integrated scattering occurred in our experiments. The $GF(85\%)$ values change by only ~1% over the range of sizes that contributed substantially to the observed scattering and thus changes in the measured GF values should only depend on composition. (To the extent that composition depends on size, size will play a role. But the Kelvin effect can be ignored for these experiments.)

[Figure]

Pages 9 and 10 : calculation of GF(RH) : the underlying hypothesis for such a iterative calculation is that the chemical composition of the aerosol is homogeneous over the whole range

of sizes (independent of the particle diameter). Figure S6 does not show this. How does this impact the results ?

This simplification has no major impact on our results. The iterative approach used in our analysis can be modified to allow GF values to vary with size. If a functional form is assumed then one can allow the parameters describing this relationship to vary, as opposed to a single average value, to match the observed f(RH) values. We have actually made this model modification, assuming that GF values are a linear function of $\log(d_p)$, increasing as $d_p$ increases. Specifically, we assumed that $GF(d_p) = a + b*\log(d_p)$ with the added constraint that $GF(1800$ nm$) = 2.1$, i.e. that these particles are pure sodium chloride and that all values must be $>= 1$. It was fully possible to adjust the a and b coefficients for each day to match the calculated and observed f(RH) values. One then finds that the lines (and a and b coefficients) vary day-to-day in a manner consistent with the derived variations in the optically-weighted GF values. The results of this approach are shown in the figure below for the Outdoor MART. The reason that this approach was not adopted as the default approach is that we did not want to introduce another assumption regarding the form of the *GF* vs. diameter relationship, and thus opted for the simpler (albeit, potentially less physically realistic) approach. We now discuss this further in the manuscript in Section 2.2.2 and have added the following text.

> "It is assumed that the growth factors are size independent, namely that $GF_x = GF$ for all $d_p$. Thus, this method retrieves an effective, optically-weighted *GF* value that explains the observed influence of water uptake on light scattering for the sampled size distribution. An alternative approach was considered in which the $GF_x$ were assumed to vary with size, specifically as $GF_x = 2.1 - b \, (\log(1.8 \, \mu m) - \log(d_{p,m}))$, and where the value of $b$ was allowed to vary during the optical closure, with the condition that $GF_x \geq 1$. (This expression assumes that particles with $d_{p,m} = 1.8$ have a $GF_x = 2.1$, i.e. that of NaCl. The $GF_x$ decrease as size decreases.) The derived $b$ values exhibit a similar temporal dependence as the derived optically-weighted *GF* values. The general conclusions reached in this study are therefore independent of the assumptions made regarding the size-dependent behavior of $GF_x$. Thus, rather than introducing an uncertain functional form, the simpler assumption (namely, size-independent $GF_x$ values) is used here."

[Figure]

Page 10, lines 1ç-21 : "The measured GF(85%) for NaCl was 2.09 +/- 0.03 and for (NH4)2SO4 was 1.59 +/- 0.05, which compare very well with literature values of ~2.1 for NaCl (Cruz and Pandis, 2000; Laskina et al., 2015; Hansson et al., 1998) and ~1.55 for ammonium sulfate (Laskina et al., 2015; Wise et al., 2003)." The literature values should be reported for a given aerosol size (or size range).

The aerosol size ranges have been added to the manuscript.

Page 12 , lines 13_14 : "Uncertainty in the assumed RI value for the dry particles may explain a small fraction (<5%) of the difference." How was this assessed ? Has the chemical analysis of the aerosol been used to estimate the real RI? All hypothesis for possible discrepancies addressed in this paragraph should be detailed in the methodology section (or at least in the supplementary material). An overall uncertainty on the HGF retrievals procedure should be calculated and compared to the measured HGF variability and consequent Org frac variability, so the reader can be convinced that the measured time variations are real. The uncertainty on the calculation method should be less than the 10 to 19% decrease in HGF for the results of the paper to be significant.

The uncertainties in the measured f(RH), relative humidity, refractive index, and diameter all contribute to uncertainty in GF retrieval. A fuller discussion of the contributions of uncertainty in each term to the uncertainty in the retrieved GF values, and how this was determined, has been added to the supplementary material. The precision of the GF values for each experimental day ranged from 1.7 to 2.2% (determined as the standard deviation of the individual measurements

over each sampling period), which is far less than the 5 to 15% (in the updated manuscript the GF(85%) for pure sea salt is assumed to be 2.1 instead of 2.2) decrease in GF relative to pure, inorganic sea salt.

Page 12, lines 21-24 : "We have tested the sensitivity of the retrieval method to an 8% increase in the particle diameters. The retrieved GF values are increased by a marginal amount (0.015-0.03) when the diameters are increased, and thus such potential sizing uncertainty does not affect the main conclusions presented here" Does this mean that a particle diameter increase of 8% was actually applied to the data set ?

This statement refers only to the sensitivity tests. In the sensitivity test, the particle diameters were all increased by 8%, and then the retrieval was performed. No adjustment was applied to the "data set" beyond this sensitivity test.

Page 17, lines 10-12 : "By using a campaign-average scaling factor, it is implicitly assumed that the actual variations in $f_{SS,0.75\mu m}$ are captured by $f_{SS,avg}$, which seems reasonable given the general constancy of the size distributions over the course of each of the microcosm experiments, c.f. Fig. 2." Why would the relative stability of the size distribution shown on fig 2 insure that the non-sea salt content of the aerosol (shown to increase in the course of the experiment) evolves uniformly with size ?

The ATOFMS sea salt fractions are no longer used quantitatively, so we no longer adjust the sea salt fraction to a vacuum aerodynamic diameter 0.75 µm.

Page 17, lines 19-24 : ". . .organic volume fraction ($\varepsilon_i$) of 0.56 – 0.88 for these particle types if it is assumed that volume mixing rules apply (i.e. the Zdanovskii-StokesRobinson mixing rules (Stokes and Robinson, 1966)). Since the non-SS values range from 0.53 – 0.74, if it is assumed that the SS-type and non-SS particle types have similar size distributions, then the implied ensemble average $\varepsilon_i$ would be about 0.33 – 0.52." I understand that the first time that $\varepsilon_i$ is used it refers to the fraction of hydrophobic material in the non-SS fraction, while the second time it is used it refers to the fraction of hydrophobic material in the overall aerosol. If this is right, the same terminology should not be used for both.

We now have an estimate for the organic volume fraction ($\varepsilon_{org}$) of the SSA particles and no longer estimate the non-SS organic fraction. Thus, there is no longer a need to change the terminology.

Page 18, lines 9-12: "Both DOC and heterotrophic bacteria concentrations increased as the bloom progressed until they stabilized around the point when Chl-a concentrations had returned approximately to their pre-bloom levels, with DOC concentrations ranging from 200 to 300 µM C and heterotrophic bacteria concentrations from 1 x 106 to a peak of 1.7 x 107 mL-1 (Figure S5B)." Are those values realistic for natural seawaters ?

As noted on Page 13 Lines 5-6 in the original manuscript, the peak DOC range is somewhat larger than values typically observed for blooms in the ocean, which are only ~130 - 250 µM C (Kirchman et al., 1991; Norrman et al., 1995). Regarding heterotrophic bacteria, the range of

heterotrophic bacteria concentrations in surface ocean waters range from around 1 to $5 \times 10^6$ cells per mL (Li, 1998), which is comparable to the bacteria concentrations observed in the MART, although the peak MART concentrations exceed those in the ocean. This has been added to the manuscript.

Technical comments
Page 5, line 8 : mesocosm or microcosm ?

"Mesocosm" has been replaced with "microcosm" to keep terminology consistent.

Figure 3 (B) : description of Org not in the figure text. "the reported uncertainties for all properties is 1 sigma: : : "should be "the reported standard deviations for all properties is 1 sigma: : :" as those are not uncertainties on the measurements

This has been updated in the manuscript.

**References**

Ault, A. P., Guasco, T. L., Ryder, O. S., Baltrusaitis, J., Cuadra-Rodriguez, L. A., Collins, D. B., Ruppel, M. J., Bertram, T. H., Prather, K. A., and Grassian, V. H.: Inside versus Outside: Ion Redistribution in Nitric Acid Reacted Sea Spray Aerosol Particles as Determined by Single Particle Analysis, 135, 14528-14531, doi:10.1021/ja407117x, 2013.

Cappa, C. D., Onasch, T. B., Massoli, P., Worsnop, D. R., Bates, T. S., Cross, E. S., Davidovits, P., Hakala, J., Hayden, K. L., Jobson, B. T., Kolesar, K. R., Lack, D. A., Lerner, B. M., Li, S.-M., Mellon, D., Nuaaman, I., Olfert, J. S., Petäjä, T., Quinn, P. K., Song, C., Subramanian, R., Williams, E. J., and Zaveri, R. A.: Radiative Absorption Enhancements Due to the Mixing State of Atmospheric Black Carbon, 337, 1078-1081, doi:10.1126/science.1223447, 2012.

Cochran, R., Laskina, O., Jayarathne, T., Laskin, A., Laskin, J., Lin, P., Sultana, C. M., Moore, K. A., Cappa, C., Bertram, T., Prather, K. A., and Grassian, V. H.: Analysis of Organic Anionic Surfactants in Fine (PM2.5) and Coarse (PM10) Fractions of Freshly Emitted Sea Spray Aerosol, 50 2477-2486, doi:10.1021/acs.est.5b04053, 2016.

Frossard, A. A., Russell, L. M., Massoli, P., Bates, T. S., and Quinn, P. K.: Side-by-side comparison of four techniques explains the apparent differences in the organic composition of generated and ambient marine aerosol particles, 48, v-x, doi:10.1080/02786826.2013.879979, 2014.

Kirchman, D. L., Suzuki, Y., Garside, C., and Ducklow, H. W.: High turnover rates of dissolved organic carbon during a spring phytoplankton bloom, 352, 612-614, doi:10.1038/352612a0, 1991.

Lack, D. A., Richardson, M. S., Law, D., Langridge, J. M., Cappa, C. D., McLaughlin, R. J., and Murphy, D. M.: Aircraft Instrument for Comprehensive Characterization of Aerosol Optical Properties, Part 2: Black and Brown Carbon Absorption and Absorption Enhancement Measured with Photo Acoustic Spectroscopy, 46, 555-568, doi:10.1080/02786826.2011.645955, 2011.

Li, W. K.: Annual average abundance of heterotrophic bacteria and Synechococcus in surface ocean waters, 43, 1746-1753, doi:10.4319/lo.1998.43.7.1746, 1998.

Mochida, M., Kitamori, Y., Kawamura, K., Nojiri, Y., and Suzuki, K.: Fatty acids in the marine atmosphere: Factors governing their concentrations and evaluation of organic films on sea-salt particles, 107, AAC 1-1–AAC 1-10, 2002.

Norrman, B., Zwelfel, U. L., Hopkinson, C. S., and Brian, F.: Production and utilization of dissolved organic carbon during an experimental diatom bloom, 40, 898-907, doi:10.4319/lo.1995.40.5.0898, 1995.

Ovadnevaite, J., Ceburnis, D., Canagaratna, M., Berresheim, H., Bialek, J., Martucci, G., Worsnop, D. R., and O'Dowd, C.: On the effect of wind speed on submicron sea salt mass concentrations and source fluxes, 117, doi:10.1029/2011jd017379, 2012.

Petters, M. and Kreidenweis, S.: A single parameter representation of hygroscopic growth and cloud condensation nucleus activity, 7, 1961-1971, doi:10.5194/acp-7-1961-2007, 2007.

Quinn, P. K., Collins, D. B., Grassian, V. H., Prather, K. A., and Bates, T. S.: Chemistry and Related Properties of Freshly Emitted Sea Spray Aerosol, doi: 10.1021/cr500713g, 2015. doi:10.1021/cr500713g, 2015.

Stokes, M. D., Deane, G. B., Prather, K., Bertram, T. H., Ruppel, M. J., Ryder, O. S., Brady, J. M., and Zhao, D.: A Marine Aerosol Reference Tank system as a breaking wave analogue for the production of foam and sea-spray aerosols, Atmos. Meas. Tech., 6, 1085-1094, doi:10.5194/amt-6-1085-2013, 2013.

Sultana, C. M., Collins, D. B., Lee, C., Axson, J. L., Santander, M. V., and Prather, K. A.: Exploration of the Effect of Biological Activity on the Chemical Mixing State of Sea Spray Aerosols During a Laboratory Phytoplankton Bloom, In Prep., In Prep.

---

## Author Comment (AC2) · 2 Jun 2016

**Author's response to reviewer's comments**

We thank the two reviewers for their constructive comments and suggestions, which have helped to improve the manuscript. Before responding to the specific individual comments from the reviewers, we note that, we have made substantial changes to the manuscript based on the reviewer comments. Specifically, we changed the focus from looking at the relationship between sea spray aerosol particle hygroscopicity and ATOFMS cluster-type fractions to one between hygroscopicity and organic matter volume fractions ($\varepsilon_{org}$). The OM volume fractions were estimated from the AMS organic matter/PM$_1$ mass fractions that were presented in the original manuscript. In the original manuscript, we did not use the $\varepsilon_{org}$ quantitatively, as there are concerns regarding the detection efficiency of the AMS for these marine derived organics as particles containing a large fraction of sea salt have a higher susceptibility to particle bounce and organic matter contained in these particles may be inefficiently vaporized (as is suggested by the results presented by Frossard et al. (2014)). That said, in one of the references mentioned by Reviewer #2 (Ovadnevaite et al. (2012)), it was determined that sea salt aerosol had a collection efficiency (*CE*) in the AMS of 0.25. We have therefore now corrected the AMS organic matter/PM$_1$ mass fractions using a *CE* of 0.25, and the mass fractions were converted to $\varepsilon_{org}$ assuming a density of 1 g/cm$^3$ for organic matter. The resulting $\varepsilon_{org}$ are therefore relatively uncertain in terms of absolute magnitude, but the trends with time should be reasonably robust under the assumption that the *CE* did not change substantially across the measurement campaign. A thorough discussion of the uncertainties in $\varepsilon_{org}$ as estimated from AMS organic matter/PM$_1$ mass fractions and the details for calculating $\varepsilon_{org}$ has been added to section 2.2.1.

> "It is important to note that while the temporal trends of the AMS NR-OM/PM$_1$ fractions are likely reflective of the general behavior, the absolute values are more difficult to quantify because NR-OM associated with particles containing high sea salt fractions may not be vaporized efficiently by the AMS due to the refractory nature of sea salt (Frossard et al., 2014) and to the susceptibility of SSA particles to particle "bounce" in the AMS. Consequently, the SSA particles, including the NR-OM component, are detected with a collection efficiency (*CE*) lower than unity (Frossard et al., 2014). One previous study (Ovadnevaite et al., 2012) determined the *CE* value for organic-free sea salt sampled when RH < 70% is approximately 0.25. However, they also note that the *CE* is potentially instrument dependent, and further may not be applicable to the organic fraction in sea spray particles due to differences in ionization efficiency (which is a component of the overall *CE*) (Ovadnevaite et al., 2012). It is also possible that the *CE* differs between particles that have differing relative amounts of OM and sea salt. Despite such uncertainties in quantification of NR-OM by the AMS for sea spray particles, the NR-OM mass concentrations for the sampled SSA particles were determined in this study assuming *CE* = 0.25. The measured NR-OM mass concentrations were used to calculate NR-OM volume concentrations assuming a density ($\rho$) of 1.0 g/cm$^3$. A value of 1.0 g/cm$^3$ for $\rho_{OM}$ is consistent with that of fatty acids ($\rho < 1$ g/cm$^3$), which are a significant fraction of marine-derived OM (Mochida et al., 2002; Cochran et al., 2016). However, this value serves as a lower bound for $\rho_{OM}$ because OM with higher densities, such as sugars ($\rho \sim 1.7$ g/cm$^3$), have also been observed in SSA (Quinn et al., 2015). The NR-OM volume

fractions of SSA ($\varepsilon_{org}$) were calculated as the ratio between the observed NR-OM volume concentrations and the integrated total particle volume concentrations from the size distribution measurements. Given the use of a lower-limit value for $\rho_{OM}$ the $\varepsilon_{org}$ are likely upper limits (not accounting for uncertainty in the assumed *CE*)."

The CE-corrected $\varepsilon_{org}$ are now used as the primary compositional metric for understanding both the depression in *GF*(85%) values relative to inorganic sea salt and their temporal variability. Figures 3 and 5 have been updated to show the CE-corrected $\varepsilon_{org}$ values. Discussion regarding the temporal variability in and absolute magnitude of the CE-corrected $\varepsilon_{org}$ has been added.

> "The NR-OM volume fractions of SSA varied from 0.29 to 0.50 throughout the course of the indoor MART microcosm experiment (Figure 3). The observation of such large $\varepsilon_{org}$ values is consistent with the substantial depressions in the *GF*(85%) values relative to pure, inorganic sea salt (2.1). The temporal variation in the $\varepsilon_{org}$ was generally similar to that of the *GF*(85%) values, with smaller *GF*(85%) values corresponding to larger $\varepsilon_{org}$ values, although the peak in $\varepsilon_{org}$ is somewhat sharper than the dip in the *GF*(85%). The inverse relationship between the *GF*(85%) and $\varepsilon_{org}$ is consistent with organic compounds being less hygroscopic than sea salt."

The original Figure 4, which showed the relationship between the *GF*(85%) values and the ATOFMS non-sea salt cluster fractions, has been replaced. The new Fig. 4 now shows the relationship between *GF*(85%) and the CE-corrected $\varepsilon_{org}$. We now use the Zdanovskii-Stokes-Robinson (ZSR) mixing rules to estimate a *GF*(85%) value for the organic matter component of the SSA particles specifically. The fitting procedure is described on Page 16 Lines 12-20 in the updated manuscript.

There has also been an evolution in our understanding of the ATOFMS clusters determined for nascent sea spray since the manuscript was originally submitted. A portion of the non-sea salt (sodium-depleted) clusters can be explained by the incomplete ionization of sea salt particles (Sultana et al., In Prep.). This change in our understanding of ATOFMS cluster types further supports our decision to use the CE-corrected $\varepsilon_{org}$ from the AMS in place of ATOFMS cluster types to understand the dependence of the *GF*(85%) on variations in particle composition. A brief discussion of this new understanding regarding the ATOFMS clusters for nascent sea spray has been added to the manuscript in the methods section.

> "It is important to note, however, that dried SSA particles sampled by the ATOFMS can be spatially chemically heterogeneous, with shells depleted in Na and rich in Mg, K, and Ca (Ault et al., 2013). Thus, some fraction of the particles identified as having Mg or SSOC type spectra may be partially explained by the incomplete ionization of sea salt particles (Sultana et al., In Prep.). However, variations in the thickness of this Na-depleted shell likely reflect variations in the total particle organic content. Therefore, increases in the fraction of SSOC or Mg type mass spectra generated suggest a net increase in SSA particle organic content."

We made other changes, where deemed appropriate and added additional figures and tables as supplemental materials. Our point-to-point response to both reviewers follows below.

**Key: Black = Reviewer, Blue = Response**

**Response to Reviewer #2**

The paper by Forestieri et al. reports on hygroscopicity of sea spray particles generated in lab conditions during various stages of phytoplankton bloom development. Lab generated sea spray studies are being pursued by many research groups during recent years trying to uncover the mechanisms and impacts of organic matter enrichment in sea spray particles. The hygroscopic properties of sea spray were studied by measuring scattering properties of wet versus dry particles. As it measures bulk sea spray population it is missing on the important aspect of size dependent chemical composition which is critical in uncovering organic matter enrichment processes. The results of the study are not particularly new and the authors could increase its significance by assessing radiative forcing impacts.

Reductions in hygroscopicity have indeed been linked to SSA particle composition changes during phytoplankton blooms in the ocean during field studies, but (to our knowledge) this is the first time this has been quantified for particles produced during a phytoplankton bloom that is completely isolated from anthropogenic influence or background particles. Thus, we believe that this work does provide a new contribution to the literature, as noted on Page 3, Lines 19-20.

The goal of this study was not to understand the size dependence of SSA particle composition, but rather to link composition to optically weighted hygroscopic growth factors for submicron particles. A figure showing the observed f(RH) relative to the f(RH) of pure sea salt for duration of both microcosm experiments has been added to supplementary material as a complement to the radiative impacts discussion in Section 4.

Finally, regarding a broader assessment of the radiative forcing impacts, although we agree with the reviewer that this would be an interesting extension, it would clearly require doing something like implementing a new scheme in a climate model and running that climate model, which is far outside the scope of this work.

It would be very interesting how the results of this study compare with the study by Vaishya et al. (2013) conducted in marine atmosphere (the study referenced, but not discussed).

We have now added extensive discussion associated with Vaishya et al. in the "Implications and Conclusions" section. Specifically, we now write:

"This was previously suggested by the ambient measurements of Vaishya et al. (2013), who observed substantial differences in $GF(90\%)$ and $f(RH)$ values for submicron particles having very different $\varepsilon_{org}$ fractions in what were identified as clean marine air masses. (Their $GF(90\%)$ values were measured using a hygroscopic tandem DMA (HT-DMA) for size-selected particles with 35 nm $\leq d_{p,m} \leq$ 165 nm. Their $f(RH)$ values were measured for $PM_1$.) They observed that increases in $\varepsilon_{org}$ had no effect on the $GF(90\%)$ until a threshold $\varepsilon_{org}$ was reached, specifically $\varepsilon_{org} > \sim 55\%$. Below this value, they measured $GF(90\%)$ value of $\sim 2.3$, which is the expected value for pure sea salt at $RH = 90\%$. Above this value, the observed a

rapid fall off in $GF(90\%)$ to a plateau at 1.22. This reported behavior differs from that observed for nascent SSA particles sampled in the current study. Here, substantial depressions in $GF(85\%)$ (and $f(RH)$) relative to inorganic sea salt were observed when the $\varepsilon_{org}$ was only ~25%, and a co-variation between $GF(85\%)$ and $\varepsilon_{org}$ (and the ATOFMS SS spectral-type fraction) was observed. One plausible reason for this difference is that nascent (freshly-emitted) SSA particles are measured here whereas *Vaishya* et al. (2013) measured ambient particles that could be subject to photochemical processing. Secondary organic aerosol formed from gases, such as monoterpenes and isoprene, emitted from the ocean (Shaw et al., 2010) could have contributed to the NR-OM, although Vaishya et al. (2013) argue that this influence was negligible based on the literature. Emission rates of such species from the ocean and their relationship with oceanic processes are not well established. Although *Vaishya* et al. (2013) attempted to remove the influence of secondary organics in their analysis (as well as the influence of non-sea salt sulfate), it is possible that their analysis was complicated by the impacts of atmospheric processing. Another key difference is that relationship between the $GF(85\%)$ values and $\varepsilon_{org}$ observed in the current study is consistent with ZSR behavior, while Vaishya et al. reported "bistable" behavior of the $GF(90\%)$ values as a function $\varepsilon_{org}$ (i.e. the flat behavior at $\varepsilon_{org} < 55\%$ and the steep fall off above). The physical basis of this bistable behavior, and the functional form implied by their measurements, is not easily explained. Finally, the $GF(90\%)$ measurements by Vaishya et al. were made for particles with $d_{p,m} < 165$ nm, while the composition was characterized with an HR-AMS. It is possible that size mismatch between these measurements influenced their analysis. Mass-weighted size distributions were not shown by Vaishya et al. (2013), however Frossard et al. show mass-weighted size distributions for ambient particles sampled in the remote marine boundary layer that suggest that much of the organic mass is contained in particles > 165 nm. Our results clearly indicate that compositional changes to nascent SSA particles, driven by variation in physical and biochemical processes in seawater, can impact the influence of water uptake on scattering by submicron SSA even when $\varepsilon_{org} < 55\%$. The comparison with the *Vaishya et al.* (2013) measurements suggests that this initial state can be further modified through atmospheric processing."

The most confusing aspect of this study is that a significant change in hygroscopicity of sea spray particles is only loosely connected to chemical composition. AMS did not detect the amount of organic matter required to explaining the observed change in GF. While the authors speculate about the bounce and refractory nature of sea spray particles (providing no references) the published evidence is in favour of AMS being able to quantitatively measure sea spray e.g. (Allan et al., 2004; Ovadnevaite et al., 2012; Schmale et al., 2013) to mention a few.

As discussed in detail at the beginning of this document, we believe that the reviewer's suggestion of better utilizing AMS data improves understanding of the observed $GF(85\%)$ values. Despite the uncertainties in quantifying organic matter in sea spray aerosol (SSA) particles by the AMS, we estimated organic matter volume fractions ($\varepsilon_{org}$) for the particles

sampled during this study. The range of $\varepsilon_{org}$ was 0.25 to 0.50, which is consistent with the observed depressions in growth factors relative to inorganic sea salt.

Frossard et al. (2014) was provided as a reference for the refractory nature of sea salt in the original manuscript (Page 15 Lines 1).

ATOFMS results seem to correlate with the observed GF, but ATOFMS lacks quantitative estimate as its sensitivity to sea spray is rather poor. As the mixed-in organic matter in sea spray would increase ATOFMS sensitivity, the amount of non-sea-salt particles would be biased high. Also considering ATOFMS size range and MART sea spray particle size peaking at a size where ATOFMS just starting to detect particles, it appears that ATOFMS measured only a fraction of sea spray population. As it currently stands, the data do not corroborate each other.

It is true that the ATOFMS results are semi-quantitative. However, trends in the data are still informative and are indicative of changes in the chemistry of the particles. Therefore, increases in the number of SSOC type spectra are indicative of chemical changes in the particle population, specifically higher organic content. Though we cannot say quantitatively the degree of organic enrichment, we can say that it occurred. Also, not all organic matter would necessarily increase the ATOFMS sensitivity. If it was very lipid rich (with lots of hydrocarbon character) the sensitivity may have even decreased.

While the number-weighted distribution peaked at 100 nm, the CRD optically weighted GFs are most sensitive to particles with $d_{p,m}$ between 400 nm to 800 nm (see Page 11 Lines 13-14 and Figure 2) . The ATOFMS counts are maximum at a vacuum aerodynamic diameter of 1.5 µm (Figure S3), corresponding to a mobility diameter ($d_{p,m}$) of 830 nm, which is a little above this range. However, since the ATOFMS is no longer used quantitatively, it is no longer necessary to adjust ATOFMS cluster fractions to smaller sizes ($d_{va}$ = 0.75 µm) as was done in the original manuscript

Page4, Line 24. I wonder if the flow was split isokinetically (equal face velocities) between instruments sampling from MART as that could affect sampled particle sizes of individual instruments. The authors mentioned laminar conditions, but laminar conditions limit particle losses to tubing walls while isokinetic split maintains the same particle population into each sampling line.

The reviewer raises an important question about the comparability of the measurements between instruments due to differences in sampling. During these experiments, flow was not split isokinetically. The particle-laden air from the MART was sampled into a manifold. The individual instruments sampled from this manifold from one of a many "ports". The flow rate to each instrument (or group of instruments), and thus the flow from each port, varied. For example, for Group 1 the CRD + SEMS sampled a much higher flow rate (3 LPM) than the AMS and ATOFMS (~ 0.7 to 1 LPM) from the manifold. Therefore, it is possible that the instruments sampled particle populations with different sizes. It is difficult to estimate differential losses between the different ports due to flow rate differences in this configuration.

The equations describing aspiration efficiency for isoaxial sampling from an air stream typically have a form similar to:

$$\eta_{asp} = 1 + \left[\frac{U_0}{U} - 1\right][other\ terms]$$

where $\eta_{asp}$ is the aspiration efficiency, $U_0$ is the ambient gas stream velocity and $U$ is the sampling velocity (Kulkarni, Baron and Willeke, 2011). The last term in brackets ("other terms") depends on particle diameter and velocity, but we will not worry about this at this time. For the manifold system here, the effective ambient gas stream velocity is very low, and will be much lower than the sampling velocity to each individual port (due in large part to the substantial difference in size between the manifold and the sampling ports. In the limit of $U_0 \rightarrow 0$ (or more specifically, $U_0 << U$), we can see that $\eta_{asp} \rightarrow 1$. Thus, it seems reasonable to think that the particle population will not be strongly influenced by the non-isokinetic sampling conditions here and the lack of explicit isokinetic sampling did not have a substantial impact on the measurements here.

Page 5, Line 12. Peak chlorophyll concentration was mentioned as 10ug/l in the previous paragraph.

Even though the peak was 10µg/l, a concentration of 12 µg/l is consistent with MART bloom studies described in Lee et al. (2015). This line has been revised to read "Further sampling was delayed until Chlorophyll-a (Chl-a) concentrations exceeded *approximately* 12 µg L$^{-1}$"

Line 17. Was this MART reproducibility issue or else? Considering 3week duration of the whole experiment a substantial degradation of organic matter (rotting) should have occurred at ambient temperatures in excess of 25C. Was bacteria growth monitored to inform on such process and if not informative, how could that be related to real world environment?

The reason for the greater contribution of larger particles in the pre-nutrient size distribution is unknown. It may have been due to differences in water level (water was collected for offline sampling once per day) or sampling tube length.

Bacterial growth in the bulk water was indeed monitored over the course of the 2 week (not 3 week) duration of each individual MART experiment. The time-series of bacterial concentrations was shown in the original manuscript in Figure S6 and was observed to peak after the chlorophyll peak. The method for this measurement has been added to Table 1. The impact of bacteria in the source water on the chemical nature and fraction of organic matter in nascent SSA particles has been discussed in more detail in Wang et al. (2015).

Page 6, Line 27. Were the particles dried? What RH? It seems that APS density was picked based on OM fractional contribution which suggests about 30% depending on OM density. If particles were not dried the picked density would not apply.

As stated in the original manuscript Page 6 Line 13, particles were dried (RH < 20%) prior to sizing. For all instruments mentioned, we stated that they measured "dried" particles.

Page 7, Line 26. Was PM2.5 cyclone operated in dry or wet conditions which could have converted PM2.5 into PM1 or lower size cut if wet?

The PM2.5 cyclone was located prior to the drier and thus operated in "wet" conditions. The RH at the point of the cyclone was around 70%, although this was not constantly monitored. The equivalent size cut for the dried particles was therefore smaller (as suggested by the reviewer), we estimate by ~1.5x (based on our derived GF values). A figure showing the cavity ring-down and SEMS sampling configuration following the manifold has been added to supplementary material (Figure S1). The RH for cyclone sampling has also been added to the manuscript (Page 8 Line 29 and Page 9 Lines 1-4).

Page 8, Line 3. Following the paragraph above referring to minimal contribution of >2.5um particles to the total SSA population it follows that ATOFMS sampled minor fraction of particles considering its transmission efficiency. Given low ATOFMS sensitivity to sea salt particles it transpires that ATOFMS sampled fraction of a fraction of SSA population. This aspect has to be clearly articulated otherwise references to SSA chemical composition is heavily biased towards supermicron particles.

In the original manuscript, the ATOFMS size-dependent counts were shown in Figure S2. As stated in the manuscript (Page 8 Line 17-18) and indicated by this figure, the peak in the particle counts for the ATOFMS is at 1.5 μm vacuum aerodynamic diameter ($d_{va}$). The particle counts fall off rapidly below $d_{va} = 0.5$ μm. As discussed in the manuscript, to relate the ATOFMS measurements to the optical property and hygroscopicity measurements requires converting the vacuum aerodynamic diameters into mobility-equivalent diameters ($d_m$). A value of $d_{va}$ 1.5 μm corresponds to a mobility-equivalent diameter of 830 nm. This value of $d_m$ is in the upper end of the size range of the particles that most contributed to the observed scattering. However, since we no longer use ATOFMS data in a quantitative way and mainly use this data for understanding temporal changes, it is no longer necessary to adjust the ATOFMS fractions to a more relevant size.

Page 9, Line 2. Is it referred to dry of wet particles? If SEMS was dried, but AMS was not then not same SSA population was measured by the two instruments making diameter match irrelevant. Wet particles entering the AMS inlet are instantly frozen due to adiabatic expansion and segregated by aerodynamic lenses based on their wet diameter. Assuming RH in the MART and subsequent sampling lines 90-100%, wet particle diameter was 2-3 times larger than dry SEMS particles. NR-OM mass was therefore limited to 186-280nm instead of 560nm. The drying issue appears quite central throughout the manuscript, so I suggest it clarifying at the beginning and using notations d(dry), d(wet) were appropriate. If AMS sampled wet particles that would explain the missing mass discussed few lines below.

As stated on Page 8 Line 18 in the original manuscript, the AMS sampled dried particles. In fact, all of the instruments used in this study ultimately sampled dried particles. Thus, instead of adopting the notation suggested above throughout the manuscript (since wet particle diameters are only discussed in the Instrumentation section), clarification has been added to specify whether particles were dried prior to sampling for each instrument in Table 1.

Also, we should correct the misconception that the RH in the MART and sampling lines was necessarily 90-100%. The RH in the MART is dictated by the balance between the evaporation rate of water and the flow rate and RH of the sampling airstream. The RH in our sampling lines was, in fact, closer to 70% RH and not 90-100%.

Line 8. AMS is typically calibrated with dry NH4NO3 particles. Why would SS particles bounce more than the calibration particles as AMS chemical species mass is calculated on nitrate equivalent basis?

Particle bounce is not significant for ammonium nitrate because it primarily exists in the liquid phase and thus has a high collection efficiency (CE) of ~100%. On the other hand, solid particles have lower CE values due to particle bounce. Issues of collection bounce for different materials and as a function of phase have been previously addressed by [Matthews et al. 2008]. Sea salt specifically has a CE of 0.25, although may vary by instrument and can depend on the extent of drying (Ovadnevaite et al., 2012).

Page 11, Line 20. Many lab and ambient studies reported chemical composition dependence on particle size which would make GF size dependent too. This study reports size independent (averaged) GF which is rather misleading and, therefore, the issue should be clearly stated.

The reported GF values here are defined as "optically weighted" to indicate just what the reviewer implies, namely that they are an average over different sizes but weighted by the scattering. To further clarify, in the abstract on Page 1 Line 22, "bulk average" has been changed to "optically-weighted average." We have also added text to Section 2.2.1 where the f(RH) to GF inversion procedure was discussed to make this issue clearer. We now state:

> "Unlike *f*(RH), *GF* values are independent of the dry particle size (above about 100 nm diameter) for particles of a given composition. Thus, variations in the optically-weighted *GF* values are driven only by variations in particle composition, specifically variations in the average composition of particles in the size range over which the optical measurements are most sensitive. For the measurements here, the sensitive size range is between about 400 nm and 800 nm with particles below 200 nm contributing almost zero to the observed scattering (see Section 3.1 below). SSA particle composition can vary with size (e.g. O'Dowd et al., 2004), and thus the *GF* itself may vary with size. The optically-weighted *GF* averages across such size-dependent variations in composition to focus on the chemical changes that most influence water uptake by the particles that most contribute to light scattering."

Line 28. The discrepancy can be partly due to shallow cut-off function of PM2.5 cyclone. Another source of discrepancy can be due to losses of wet particles and corresponding losses in dryers as in general wet particles are lossier. Again the drying of the particle is very unclear throughout the study and difficult to interpret.

We have now clarified the experimental configuration and the drying aspects within the manuscript. In very general terms, all instruments used in this study ultimately sampled dried particles. However, the particles that passed through the cyclone were not dried, but at an RH ~

70%; they were subsequently dried prior to sampling. Below is a figure showing the general sampling configuration, and has been added to the supplementary material (Figure S1).

Although the reviewer is correct to note that losses increase with size, it is important to realize that the effect of water uptake on particle losses is not straightforward. Water is typically less dense (1 g cm$^{-3}$) than many other common atmospheric materials. Thus, if water uptake leads to a decrease in density then this can offset, at least to some extent, the increase in size in terms of sedimentation losses. Consider an example. If a 1 micron particle has a density of 2.0 g cm$^{-3}$ (e.g. sea salt) and doubles in size due to water uptake (e.g. GF = 2.0) then the density of the particle will decrease to 1.22 g cm$^{-3}$. The percent loss of a 1 micron particle with density = 2.0 g cm$^{-3}$ due to sedimentation in a 10 m long tube at a flow rate of 5 lpm is 9.3%. If the particle size were doubled without changing the density, the loss would increase to 32%. But, if the decrease in density is accounted for the loss only increases to 13%. Thus, the decrease in density offsets a very large fraction of the increase in size. (The above calculations were performed using the Particle Loss Calculator of Von der Weiden et al. (2009).) Now, of course, if the density of the material were closer to water (such as may be the case for organics) this offsetting effect would be smaller. But our premise is that the organic material is relatively non-hygroscopic, and thus the water uptake itself (and associated increase in size) would be smaller, negating the effect in the first place. Consequently, while it is possible that differences in the influence of sedimentation between instruments due to differences in drying, the magnitude of the difference is much smaller than one might intuit based on the size change alone. As such, while it may be possible that the optical closure may have been impacted by differential sedimentation of wet and dry particles such and impact is limited in scope. Finally, we note that the driers used to dry particles sampled into the CRD-PAS and SEMS were oriented vertically (to minimize sedimentation losses).

[Figure]

Figure S1. A detailed schematic of the general sampling scheme for the online instruments. Note that not all instruments sampled at the same time (see Table 1). Particles sampled from the MART passed through a manifold from which they were subsampled to the various instrumentation. All instruments included an upstream drier and sampled dried particles. The driers and humidifiers for the CRD and SEMS sampling group (Group 1) were oriented

vertically. The particles sampled to the CRD and SEMS alternately passed through a PM$_{2.5}$ cyclone. The RH at this point was ~70%.

Page 12, Line 6. Wiedensohler et al. (2012) reported that in general sizing errors of different instruments can be objectively up to 10%.

The reviewer is correct that the sizing errors of different instruments can be objectively up to 10%. In our case, we characterized the sizing accuracy of the SEMS using size-selected PSLs and found that the particle sizes were characterized to within 1% of the stated PSL size (see Page 12 Lines 7-12 in the original manuscript). Thus, it seems unlikely that large instrumental sizing errors are the primary reason for differences between the observed and calculated dry particle scattering. However, we have added a reference to the Wiedensohler paper to the discussion on Page 12 as motivation for considering the possibility of sizing errors. "*Wiedensohler et al. (2012) reported that sizing errors between instruments can be up to 10%.*"

Page 13, Line 13. 2.2 at 85% or 90%? Also on page 10, GF(85%) of NaCl was referred to as 2.1.

A *GF*(85%) of 2.2 was originally used as a value for sea salt and not NaCl. However, a value of 2.1 is a better estimate for sea salt (Ming and Russell, 2001) at 85% relative humidity and the manuscript has been updated using this value for sea salt.

Line 27. Is it possible that the relative abundance of Fe-rich particles was due to higher sensitivity of ATOFMS to Fe-rich versus SSA?

The reviewer raises a good point about differential sensitivity of the ATOFMS. However, here we are confident that the higher relative abundance of Fe-rich particles at the beginning of the experiments is due to the addition of iron rich nutrients to encourage phytoplankton growth. As the reviewer notes, the ATOFMS is very sensitive to iron compared to species such as sodium chloride making exact quantification difficult. However, we emphasize that the *trends* in the particle spectra and the particle type abundances are reflective of changes in the particle composition. During many other "microcosm" experiments using the same methodology, an initial spike in iron signal after nutrient addition was observed, which then declines as the bloom progresses and nutrients are likely taken up into the proliferating microbiology. Finally, we note that while differences in sensitivity between particle types would certainly give rise to errors in particle type quantification (relative abundance) at a given point in time, it should not give rise to time-dependent changes in the relative abundance.

Page 14, Line 23. This is only true if ATOFMS and CRD size ranges were exactly the same which was not the case as ATOFMS cannot reliably detect 100nm particles, especially SSA.

The reviewer again raises a good point about particle size and comparability. Here, we reemphasize that we have measured optically-weighted growth factors and, for the size distributions from the MART, particles with $d_{p,m} <= 100$ nm contributed very little to overall scattering measured by the CRD. As stated on Page 11 Lines 16-20 in the original manuscript, the optically-weighted GF measurements were most sensitive to $d_{p,m}$ (mobility diameters)

between 400 to 800 nm, with median scattering occurring at $d_{p,m}$ = 530 nm. As such, we compared the optically-weigthed GF values to the ATOFMS composition for particle having $d_{va}$ ~ 0.75 μm, which corresponds to a $d_{p,m}$ = 420 nm.

Page 16, Line 13. Page 10 referred to 2.1 GF(85%). Why GF=1 is expected as the minimum combined value? Any reference to backup? Marine gels and micelles have been reported to process some water despite being generally hydrophobic (Ellison et al., 1999; Chakraborty and Zachariah, 2007). Fatty acid is only one of the many possible compounds and necessarily entirely hydrophobic.

While it is true that many types of marine organic matter have GF>1, we assume a GF=1 as a lower bound. Although a lower bound, this assumption is consistent with GFs for fatty acids often found in marine aerosols (e.g. Cochrane et al., 2016). That said, we have revised the sentence to read (added text in italics): "The line connecting $GF_{SS}$(85%) = 2.2 and $GF_{non-SS}$(85%) = 1.0 provides the minimum value *(lower bound)* expected for any combination of SS and non-SS particles."

Line 20. It has been demonstrated in numerous studies that OM fraction in sea spray is size dependent. Should the GF value of 1.39 be interpreted as a bulk average of highly enriched and poorly enriched SS particles?

As noted above, we are now using $\varepsilon_{org}$ instead of the fraction of ATOFMS non-SS particles to examine the relationship between particle hygroscopicity, $GF$(85%), and composition. Therefore, we now estimate GF values for the organic fraction of the sampled PM specifically, i.e. the $GF$(85%) values after extrapolation of our fits to $\varepsilon_{org}$ = 1. We find values of $GF_{org}$(85%) = 1.16 and 1.23 for the two MART experiments, which are optically weighted averages. One can assume that the GF is constant with size or that it varies with size, perhaps with an inverse relationship between GF and size, since OM fraction typically increases with decreasing size, within the optically-weighted size range. As discussed in detail in response to Reviewer #1, it is fully possible to assume some relationship between particle size and GF to come up with an optically-weighted average value. To clarify, we have added the following sentence: "This value for $GF_{org}$ can be interpreted as an optically-weighted average for the OM component of the SSA particles sampled here."

Page 17, Line 20. There is an issue regarding size dependent chemical composition. As scattering is dominated by larger submicron sizes and the smaller submicron particles tend to be more enriched in OM, averaged GF of this study missing out on the important aspect of size dependent chemical composition.

We do not dispute that the GF may be size dependent (O'Dowd et al., 2004; Prather et al., 2013). However, we emphasize again that we have measured the *optically-weighted average GF*. Thus, the measured optically-weighted GF is directly relevant to the actual impact of composition variations on SSA particle light scattering. Put another way, composition changes of e.g. 50 nm particles are almost completely irrelevant to the magnitude of light scattering by SSA particles and the direct effect (although critical to understanding the impact of SSA particles on clouds via their ability to act as CCN). It is instead variations in the average composition within the

optically-relevant range, which is around 400-800 nm in this study given the size distribution, that is most important to consider when considering the total scattering. Had we instead measured size-dependent GF values explicitly, this would have provided, perhaps, greater process-level information. However, without considering which particles in the size distribution do most of the scattering, such process-level information is limited in nature. Since the MART system generates particle size distributions that are very similar to those generated from real wave breaking (Prather et al., 2012; Stokes et al., 2014), our optically-weighted measurements are of direct relevance to understanding the impact that compositional variations have on overall light scattering. As noted above, we have revised Section 2.2.1 to more clearly address this issue of size-dependent composition.

Line 24. How this volume fraction compared with AMS chemical composition? Did AMS record any substantial organics as 0.33-0.52 volume fraction would suggest? Figures show that AMS OM fraction was 0.05.

The reviewer raises an important question about comparability between the derived organic volume fractions and the AMS measurements. As stated at the beginning of our responses, we estimated $\varepsilon_{org}$ values from the AMS measurements instead of deriving organic volume fractions from our *GF* values. The estimated GFs for the organic component are above 1 indicating that our estimates for $\varepsilon_{org}$ are reasonable.

Page 18, Line 5. "which was 5 times higher". Much higher chl was probably due to higher temperature than the ocean (what was the T range?) and plentiful nutrients.

In the manuscript, we attribute the larger Chl-a concentrations to larger photosynthetically active radiation. However, temperature and nutrients could have also played a role. The temperature of the water in the MART was ~26°C, which is larger than the 20 to 23°C measured for seawater at the Scripps pier (Table 2).

'Page 19, Line 3. Consider different size ranges sampled if AMS was not dried. Table 1. AMS size range is missing.

As stated above, particles sampled by the AMS were dried. The size range is now included in Table 1.

**References**

Ault, A. P., Guasco, T. L., Ryder, O. S., Baltrusaitis, J., Cuadra-Rodriguez, L. A., Collins, D. B., Ruppel, M. J., Bertram, T. H., Prather, K. A., and Grassian, V. H.: Inside versus Outside: Ion Redistribution in Nitric Acid Reacted Sea Spray Aerosol Particles as Determined by Single Particle Analysis, 135, 14528-14531, doi:10.1021/ja407117x, 2013.
Cochran, R., Laskina, O., Jayarathne, T., Laskin, A., Laskin, J., Lin, P., Sultana, C. M., Moore, K. A., Cappa, C., Bertram, T., Prather, K. A., and Grassian, V. H.: Analysis of Organic Anionic Surfactants in Fine (PM2.5) and Coarse (PM10) Fractions of Freshly Emitted Sea Spray Aerosol, 50 2477-2486, doi:10.1021/acs.est.5b04053, 2016.
Frossard, A. A., Russell, L. M., Massoli, P., Bates, T. S., and Quinn, P. K.: Side-by-side comparison of four techniques explains the apparent differences in the organic composition of

generated and ambient marine aerosol particles, 48, v-x, doi:10.1080/02786826.2013.879979, 2014.

Lee, C., Sultana, C. M., Collins, D. B., Santander, M. V., Axson, J. L., Malfatti, F., Cornwell, G. C., Grandquist, J. R., Deane, G. B., Stokes, M. D., Azam, F., Grassian, V. H., and Prather, K. A.: Advancing Model Systems for Fundamental Laboratory Studies of Sea Spray Aerosol using the Microbial Loop, 119 8860–8870, doi:10.1021/acs.jpca.5b03488, 2015.

Ming, Y. and Russell, L. M.: Predicted hygroscopic growth of sea salt aerosol, 106, 28259-28274, doi:10.1029/2001JD000454, 2001.

Mochida, M., Kitamori, Y., Kawamura, K., Nojiri, Y., and Suzuki, K.: Fatty acids in the marine atmosphere: Factors governing their concentrations and evaluation of organic films on sea-salt particles, 107, AAC 1-1–AAC 1-10, 2002.

O'Dowd, C. D., Facchini, M. C., Cavalli, F., Ceburnis, D., Mircea, M., Decesari, S., Fuzzi, S., Yoon, Y. J., and Putaud, J.-P.: Biogenically driven organic contribution to marine aerosol, 431, 676-680, doi:10.1038/nature02959, 2004.

Ovadnevaite, J., Ceburnis, D., Canagaratna, M., Berresheim, H., Bialek, J., Martucci, G., Worsnop, D. R., and O'Dowd, C.: On the effect of wind speed on submicron sea salt mass concentrations and source fluxes, 117, doi:10.1029/2011jd017379, 2012.

Prather, K. A., Bertram, T. H., Grassian, V. H., Deane, G. B., Stokes, M. D., DeMott, P. J., Aluwihare, L. I., Palenik, B. P., Azam, F., and Seinfeld, J. H.: Bringing the ocean into the laboratory to probe the chemical complexity of sea spray aerosol, 110, 7550-7555, doi:10.1073/pnas.1300262110, 2013.

Quinn, P. K., Collins, D. B., Grassian, V. H., Prather, K. A., and Bates, T. S.: Chemistry and Related Properties of Freshly Emitted Sea Spray Aerosol, doi: 10.1021/cr500713g, 2015. doi:10.1021/cr500713g, 2015.

Shaw, S. L., Gantt, B., and Meskhidze, N.: Production and Emissions of Marine Isoprene and Monoterpenes: A Review, Adv. Meteorol., 2010, doi:10.1155/2010/408696, 2010.

Sultana, C. M., Collins, D. B., Lee, C., Axson, J. L., Santander, M. V., and Prather, K. A.: Exploration of the Effect of Biological Activity on the Chemical Mixing State of Sea Spray Aerosols During a Laboratory Phytoplankton Bloom, In Prep., In Prep.

Von der Weiden, S., Drewnick, F., and Borrmann, S.: Particle Loss Calculator–a new software tool for the assessment of the performance of aerosol inlet systems, 2, 479-494, 2009.

Wang, X., Sultana, C. M., Trueblood, J., Hill, T. C. J., Malfatti, F., Lee, C., Laskina, O., Moore, K. A., Beall, C. M., McCluskey, C. S., Cornwell, G. C., Zhou, Y., Cox, J. L., Pendergraft, M. A., Santander, M. V., Bertram, T. H., Cappa, C. D., Azam, F., DeMott, P. J., Grassian, V. H., and Prather, K. A.: Microbial Control of Sea Spray Aerosol Composition: A Tale of Two Blooms, 1, 124-131, doi:10.1021/acscentsci.5b00148, 2015.

---

## Author Response (AR2)

**Revised Response to Reviewer #2 Comment**

Dear Prof. Knopf,

We thank you for catching the error in our discussion of the aspiration efficiency and the potential impact of non-isokinetic sampling. We provide below a revised response to the reviewer comment (with more explicit calculations). We have also added a very short discussion to the main text (given below). Although our initial presentation of the aspiration efficiency was in error, we note that our conclusion was correct, namely that the differences in sampling between instrument groups from the manifold and the lack of explicit isokinetic sampling did not lead to the instrument groups sampling different particle populations.

Sara Forestieri and Chris Cappa

**Authors' Comment:** Page4, Line 24. I wonder if the flow was split isokinetically (equal face velocities) between instruments sampling from MART as that could affect sampled particle sizes of individual instruments. The authors mentioned laminar conditions, but laminar conditions limit particle losses to tubing walls while isokinetic split maintains the same particle population into each sampling line.

**Revised Response:** The reviewer raises an important question about the comparability of the measurements between instruments due to differences in sampling. During these experiments, flow was not split isokinetically. The particle-laden air from the MART was sampled into a manifold. The individual instruments sampled from this manifold from one of a many "ports". The flow rate to each instrument (or group of instruments), and thus the flow from each port, varied. For example, for Group 1 the CRD + SEMS sampled a much higher flow rate (3 LPM) than the AMS and ATOFMS (~ 0.7 to 1 LPM) from the manifold. Therefore, it is possible that the instruments sampled particle populations with different sizes. It is difficult to estimate differential losses between the different ports due to flow rate differences in this configuration. The equations describing aspiration efficiency for isoaxial sampling from an air stream typically have a form similar to (Kulkarni et al., 2011):

$$\eta_{asp} = 1 + \left[\frac{U_0}{U} - 1\right]\left[1 - \frac{1}{1 + kStk}\right]$$

where $\eta_{asp}$ is the aspiration efficiency, $U_0$ is the ambient gas stream velocity, $U$ is the sampling velocity, $k = 2 + 0.67(U/U_0)$ and Stk is the Stokes number, where

$$Stk = \frac{\tau_p \cdot U_0}{D_{tubing}}$$

5    with $\tau_p$ the (size-dependent) particle relaxation time and $D_{tubing}$ the diameter of the sampling tube. For the manifold system here, the effective ambient gas stream velocity in the sampling manifold was 0.08 m/min, while the CRD and the AMS had sampling velocities of 26.5 and 8.8 m/min, respectively and the inlet diameters on the manifold were ~0.012 meters for both the CRD and AMS. For particles < 1 μm (particle relaxation times ~$10^{-6}$ s), the aspiration efficiencies were

10    calculated to be nearly 100% for both systems. For particles > 10 μm (particle relaxation time ~$10^{-4}$ s) , the aspiration efficiencies were 88% and 96% for the CRD and AMS, respectively; such large particles make up a negligible fraction of the total particle populations sampled here. Thus, the particle population will not be strongly influenced by the non-isokinetic sampling conditions and the lack of explicit isokinetic sampling did not have a substantial impact on the measurements

15    here. We have added the following text to the manuscript associated with this issue:

[revised manuscript text omitted]

The supplementary material consists of six figures that provide additional support for the conclusions presented in the paper.

Table S1. Summary of uncertainties for growth factor (GF) retrieval.

| Parameter | Default Value | Perturbation | ΔGF | % ΔGF |
|---|---|---|---|---|
| f(RH) | 3.7 | 0.5 (7%)[#] | 0.05 | 2.6% |
| Relative Humidity | 85% | 1.2%/0.03% (see Table 2) | 0.05/0.01[$] | 2.4%/0.5% |
| Refractive Index | 1.55 | 0.04 | 0.05 | 2.7% |
| Particle Diameter | Distribution with mode = ~112 nm | +1% | 0.01 | <1% |

[$] ΔGF values were calculated using Kappa-Kohler equation and assuming a κ value of 1.3 [*Petters and Kreidenweis*, 2007].

Table S2. Measured and actual GFs for pure substances and the implied error in the measured RH.

| | Observed | Actual | ΔRH |
|---|---|---|---|
| NaCl | 2.09 | 2.1 | 0.03% |
| Ammonium Sulfate | 1.59 | 1.55 | 1.2% |

Uncertainties in the measured f(RH), relative humidity, refractive index, and diameter that contribute to the overall uncertainty in GF retrieval are provided in Table S1. The uncertainty in the CRD extinction is ~5% at 532 nm and the fundamental performance of the CRD method for wet particles is not changed. Therefore, the propagated uncertainty of f(RH) (=$b_{ext,wet}/b_{ext,dry}$) = $\sqrt{0.05^2 + 0.05^2}$ = 7%. Two estimates for uncertainty in relative humidity were based on the hygroscopic growth factors of pure NaCl and pure ammonium sulfate generated from a TSI atomizer. The measured values were compared to literature values to infer the error in RH (see Table 2). The refractive index used in this study is appropriate for NaCl. However, refractive index of sea salt mixed with marine derived organic matter is not well known, but a value of 1.48 reported by Nessler et al. [2005] for organic matter has been used in many recent studies [*Partanen et al.*, 2014; *Vaishya et al.*, 2013] for marine derived organic matter. The refractive index of the mixture

is likely to be somewhere in between. Assuming that organic matter is 50% of the particles by volume (consistent with the ensemble average fraction reported in this manuscript), the volume-weighted refractive index is 1.51. GF values were retrieved with a refractive index of 1.51 and compared to the GF values retrieved using the default value of 1.55 to assess the uncertainty in the refractive index. The uncertainty of 1% for the measured diameter was determined during the experiments in which a 2$^{nd}$ DMA size-selected particles 100-300 nm.

[Figure]

Figure S1. A detailed schematic of the general sampling scheme for the online instruments. Note that not all instruments sampled at the same time (see Table 1). Particles sampled from the MART passed through a manifold from which they were subsampled to the various instrumentation. All instruments included an upstream drier and sampled dried particles. The driers and humidifiers for the CRD and SEMS sampling group (Group 1) were oriented vertically. The particles sampled to the CRD and SEMS alternately passed through a PM$_{2.5}$ cyclone. The RH at this point was ~70%.

[Figure]

Figure S2. Dual polarity ATOFMS Mass Spectra for the major spectra categories: sea salt (SS), sea salt with organic carbon (SSOC and SSOC2), Iron (Fe), Organic (Org), and Magnesium (Mg) types.

[Figure]

Figure S3. Size-resolved ATOFMS particle counts.

[Figure]

Figure S4. Predicted particle losses for particles travelling from the MART outlet to the MART manifold for a sampling line 10' in length and 3/8" in diameter. The Particle Loss Calculator of [*Von der Weiden et al.*, 2009] was used.

[Figure]

Figure S5. Calculated extinction using SEMS size distributions (real RI = 1.55) for PM$_{2.5}$ and SEMS+APS size distributions as a function of the observed CRD extinction for the 2014 MART experiments. Slopes for linear fits (with the intercept fixed at 0) of calculated extinction as a function of observed extinction were 0.85 and 0.84 for PM$_{2.5}$ and PM$_{all}$, respectively. A 1:1 line is provided for reference.

[Figure]

Figure S6. Time series of concentrations of dissolved organic carbon (DOC; μM C), heterotrophic bacteria (#/mL), and chlorophyll-a concentrations (μg/L) in the seawater water for the (A) indoor and (B) outdoor MARTs.

[Figure]

Figure S7. AMS m/z 43 particle time of flight (pTOF) mass distributions for the indoor (blue) and outdoor (red) MARTS.

[Figure]

Figure S8. Calculated fraction of scattering relative to pure sea salt particles at 85% RH as a function of time for the two microcosm experiments.